# RNase III-CLASH of multi-drug resistant *Staphylococcus aureus* reveals a regulatory mRNA 3′UTR required for intermediate vancomycin resistance

Daniel G. Mediati[1], Julia L. Wong[1], Wei Gao[2], Stuart McKellar [3], Chi Nam Ignatius Pang[1], Sylvania Wu[1], Winton Wu[1], Brandon Sy [1], Ian R. Monk [2], Joanna M. Biazik[4], Marc R. Wilkins [1], Benjamin P. Howden [2], Timothy P. Stinear [2], Sander Granneman [3] & Jai J. Tree[1✉]

Treatment of methicillin-resistant *Staphylococcus aureus* infections is dependent on the efficacy of last-line antibiotics including vancomycin. Treatment failure is commonly linked to isolates with intermediate vancomycin resistance (termed VISA). These isolates have accumulated point mutations that collectively reduce vancomycin sensitivity, often by thickening the cell wall. Changes in regulatory small RNA expression have been correlated with antibiotic stress in VISA isolates however the functions of most RNA regulators is unknown. Here we capture RNA–RNA interactions associated with RNase III using CLASH. RNase III-CLASH uncovers hundreds of novel RNA–RNA interactions in vivo allowing functional characterisation of many sRNAs for the first time. Surprisingly, many mRNA–mRNA interactions are recovered and we find that an mRNA encoding a long 3′ untranslated region (UTR) (termed *vigR* 3′UTR) functions as a regulatory 'hub' within the RNA–RNA interaction network. We demonstrate that the *vigR* 3′UTR promotes expression of *folD* and the cell wall lytic transglycosylase *isaA* through direct mRNA–mRNA base-pairing. Deletion of the *vigR* 3′UTR re-sensitised VISA to glycopeptide treatment and both *isaA* and *vigR* 3′UTR deletions impact cell wall thickness. Our results demonstrate the utility of RNase III-CLASH and indicate that *S. aureus* uses mRNA-mRNA interactions to co-ordinate gene expression more widely than previously appreciated.

[1] School of Biotechnology and Biomolecular Sciences, University of New South Wales, Sydney, NSW, Australia. [2] Department of Microbiology and Immunology, Peter Doherty Institute, University of Melbourne, Melbourne, VIC, Australia. [3] Centre for Systems and Synthetic Biology, University of Edinburgh, Edinburgh, UK. [4] Electron Microscopy Unit, University of New South Wales, Kensington, NSW, Australia. ✉email: j.tree@unsw.edu.au

Staphylococcus aureus is a highly adaptable opportunistic pathogen capable of causing a wide range of infections. S. aureus is a leading cause of osteomyelitis, infective endocarditis, and bacteraemia. Methicillin-resistant S. aureus (MRSA) has become common in both community and healthcare settings with global prevalence ranging from 13 to 74% of S. aureus infections[1,2]. Mortality rates are high among patients that develop MRSA bacteraemia[2] and treatment options are often limited to last-line antibiotics[1]. The treatment of choice for patients with MRSA bacteraemia is the glycopeptide vancomycin[1,3]. Vancomycin resistant S. aureus (VRSA, MIC > 8 µg/mL) is relatively rare and vancomycin treatment failure is most commonly associated with S. aureus strains that have vancomycin-intermediate resistance, termed VISA (MIC 4–8 µg/mL)[3]. The genetic determinants that lead to VISA are heterogeneous and incompletely defined, although it is clear that these strains do not acquire additional genetic elements. Genome sequencing has identified single nucleotide polymorphisms (SNPs) that arise during repeated vancomycin exposure and commonly lead to cell wall thickening, reduced autolysis, and reduced acetate catabolism[3]. Cell wall thickening is thought to reduce diffusion of vancomycin to the division septum however the underlying mechanism of cell wall thickening remains unclear.

In this study, we have used the MRSA isolate JKD6009 which was isolated from a patient in New Zealand with bacteremia. Over 42 days of treatment with vancomycin, a second isolate from the patient presented a VISA phenotype (designated strain JKD6008)[4]. These two isolates belong to ST239, a sequence-type representing a global multi-drug resistant, hospital-associated S. aureus clone. The VISA isolate JKD6008 has acquired at least nine SNPs when compared to the parental vancomycin-sensitive (VSSA) isolate JKD6009, including mutations in the transcriptional regulators graRS and walKR[4]. Consistent with many VISA isolates, JKD6008 exhibited reduced autolysis and cell wall thickening.

Regulatory small RNA (sRNA) are approximately 50-500 nucleotide gene regulators that form base-paired interactions with mRNAs and inhibit or activate mRNA translation, transcription, and stability through a range of mechanisms. Using a collection of VSSA and VISA isolates, Howden et al. demonstrated that sRNA expression is correlated with exposure to last-line antibiotics, suggesting that sRNAs may constitute an acute antibiotic response[5]. More recently, Dejoies et al. identified a coordinated expression profile of specific sRNAs under acute biocide stress[6]. The transcriptome of S. aureus contains hundreds of sRNAs, most of which have unknown biological functions. The sRNA SprX represses the transcriptional regulator SpoVG and alters intermediate glycopeptide tolerance[7]. More recently, the noncoding antisense RNA, SprF1, was shown to bind ribosomes and reduce protein synthesis to favour ciprofloxacin and vancomycin persister cell development[8].

In S. aureus, the double-stranded RNA-specific endonuclease RNase III plays a major role in rRNA maturation and mRNA processing. RNase III typically binds 22-nt RNA duplexes[9] with a preference for at least one GC or CG base pair at cleavage sites[10]. RNA duplexes can be formed in cis or trans and many sRNA-mRNA pairs in S. aureus are processed by RNase III. Interstingly, RNase III also recognises a co-axially stacked interaction between RNAIII and coa mRNA that pairs through a short kissing loop interaction[11,12]. This interaction mimics an extended RNA duplex and is processed by RNase III indicating that shorter regions of complementarity may also be RNase III substrates.

In this work, we have adapted our proximity-dependant RNA ligation technique (RNase-CLASH) for profiling sRNA-mRNA interactions[13] to MRSA and captured RNA–RNA interactions associated with RNase III. RNase III-CLASH identifies RNase III binding sites enriched within the untranslated regions (UTRs) of mRNAs and recovers many RNAs that act as regulatory 'hubs'. We reveal the in vivo targets for many uncharacterised sRNAs providing insight into their function. We also utilise differential RNA-seq (dRNA-seq) and Term-seq to accurately map the 5′ and 3′ RNA boundary ends and identified many sRNAs not previously identified in S. aureus. We use this RNA interaction network to identify regulatory RNAs that may contribute to intermediate vancomycin resistance and find that a previously unknown regulatory mRNA, here termed vigR, is required for vancomycin-intermediate tolerance. The vigR mRNA encodes an unusually long 3′UTR that acts in trans to positively regulate isaA and folD mRNAs through direct base-pairing interactions. IsaA is a lytic transglycosylase involved in cell wall peptidoglycan turnover and expansion, and we demonstrate that it also plays a role in intermediate glycopeptide tolerance. Deletion and CRISPRi knockdown of isaA significantly reduces cell wall thickness of the VISA isolate JKD6008, while the vigR 3′UTR deletion has a modest reduction in cell wall thickness. Our study has uncovered a previously unknown but important mechanism of intermediate vancomycin resistance through a regulatory mRNA 3′UTR that promotes cell wall thickening in a clinical VISA isolate.

## Results

**Transcriptome architecture of methicillin-resistant Staphylococcus aureus JKD6009.** The 5′ and 3′UTRs of mRNAs are often sites of regulatory RNA interactions, but they are poorly predicted in silico. To facilitate accurate mapping of RNA–RNA interactions to genomic features and identify novel regulatory RNAs in S. aureus JKD6009 we utilised dRNA-seq[14] and Term-seq[15] to map RNA 5′ and 3′ boundaries, respectively. Total RNA was extracted from cells grown in liquid Mueller–Hinton (MH) to an $OD_{600}$ ~0.6 and sequenced using the dRNA-seq and Term-seq protocols (Supplementary Methods). Analysis of the dRNA-seq data using the ANNOgesic workflow[16] identified 1399 transcription start sites (TSS) and 17 RNA processing sites (Fig. 1a). Primary, secondary, antisense, and internal promoters were also identified (Fig. 1a) and are detailed in Supplementary Data 1. Motif analysis of our TSS identified a canonical -10 element (Pribnow box) (Supplementary Fig. 1A) and our primary 5′ ends are in agreement with previously published dRNA-seq data from the MRSA isolate USA300[17], and TSS-EMOTE analysis of isolate MW2[18] supporting the veracity of our data (Supplementary Fig. 1B). The 5′UTRs of mRNAs had a median length of 41-nts, with 281 5′UTRs (24.3%) longer than 100 nts (Supplementary Fig. 1C).

We next used Term-seq[15] to identify 3′ RNA ends. Our Term-seq analysis identified 2385 statistically significant RNA 3′ ends, including 798 that overlapped a predicted Rho-independent transcription terminator. A stem-loop RNA structure was identified within our RNA 3′ ends consistent with the presence of intrinsic terminators and stabilising stems at many of these sites[19] (Fig. 1b). Our RNA 3′ ends defined 1031 mRNA 3′UTRs that had a median length of 75-nts, and 346 3′UTRs (33.5%) that were longer than 100-nts (Supplementary Fig. 1D).

Our dRNA-seq and Term-seq data also allowed identification of novel non-coding regulatory RNA elements. Using combined RNA-seq, dRNA-seq, and Term-seq data as input for ANNOgesic[16], we identified 141 potential sRNAs which included 50 previously reported in S. aureus[20] (Supplementary Data 2). Of these predicted sRNAs, 57 had RNA 5′ and 3′ ends defined by both dRNA-seq and Term-seq (Supplementary Data 2). An additional 357 trans-encoded S. aureus sRNAs have been previously reported[20] and we included these elements in our subsequent analyses.

Collectively, the RNA 5′ and 3′ end data provide a high resolution, condition-specific map of transcript boundaries,

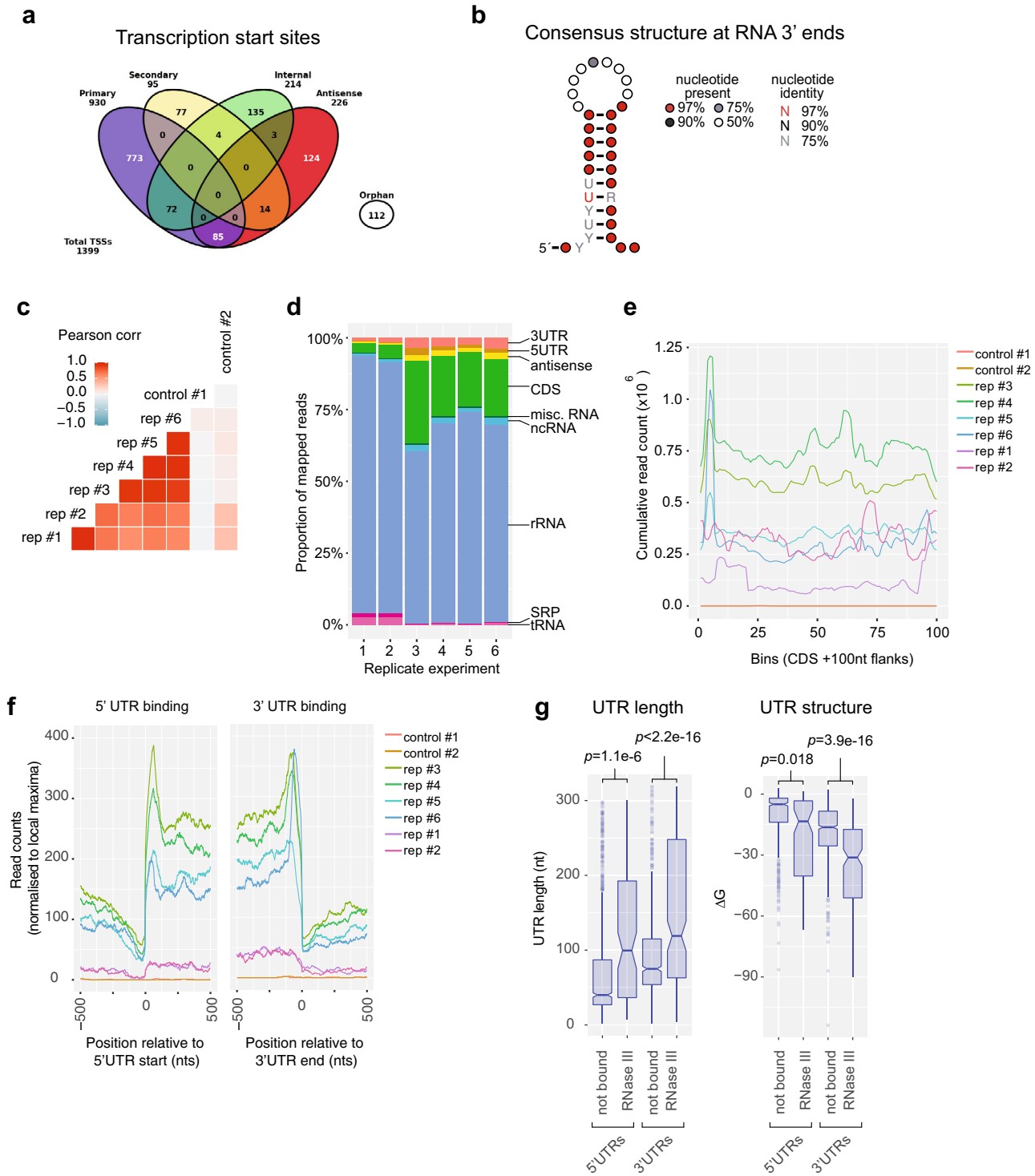

**Fig. 1 RNase III interactions with the ANNOgesic annotated *S. aureus* JKD6009 transcriptome. a** Transcription start sites and putative promoter regions identified through the ANNOgesic workflow. **b** The consensus 10-nt stem-loop RNA structure identified within our RNA 3' boundary ends. Nucleotides are represented as coloured dots specifying the nucleotide percentage identify. **c** Heatmap showing the two-sided Pearson's correlation between replicate RNase III CRAC datasets. **d** Proportion of RNase III-bound reads mapping to genomic features for replicate datasets. **e** Cumulative count of RNase III-bound reads mapping across CDSs. Each CDS was divided into 100 bins and the read depth within each bin is indicated. **f** Cumulative count of RNase III-bound reads mRNA 5'UTR ends and 3'UTR ends. Read counts for each individual UTR were normalised to 1 to prevent biases from abundant sites. The read count represents the cumulative normalised read count at 5' and 3'UTRs. **g** Distribution of RNase III-bound ($n = 104$) and not bound ($n = 1052$) UTR lengths (nts) (*left*) and structure (expressed as the free energy of the folded UTR, kcal/mol) (*right*). UTRs were classified as RNase III-bound if they contained >10 reads in two independent experiments. *p*-values were calculate using a two-sided *t*-test. For each boxplot, centre represents the median, box minima and maxima represent the 25th and 75th percentile respectively, and whiskers represent 1.5× the interquartile range.

defining mRNA 5′ and 3′UTRs, and 92 novel regulatory sRNAs in the MRSA isolate JKD6009.

**UV-crosslinking reveals RNase III bound transcripts**. In our previous analysis of the enterohaemorrhagic *E. coli* RNA inter-action network, we used the endoribonuclease RNase E (*rne*) as bait to capture sRNA–RNA interactions using an UV-crosslinking and RNA proximity-dependant ligation technique termed CLASH[13]. While *S. aureus* does not encode an orthologue of *rne*, many sRNA–mRNA pairs are substrates for the double-stranded RNA-specific endoribonuclease RNase III[11,21–25] that interacts with the sRNA–mRNA duplex. To facilitate high-stringency purification of UV-crosslinked RNA-RNase III com-plexes we created a translational fusion of the chromosomal copy of RNase III (*rnc*) with the dual affinity tag, His6-TEV-FLAG (HTF) called JKD6009 *rnc*-HTF. RNase III is not essential in *S. aureus*[26] and JKD6009 *rnc*-HTF produced wild-type levels of mature 16S and 23S ribosomal RNA (rRNA) (Supplementary Fig. 2A), demonstrating that the fusion protein is functional. IMAC purification of RNase III-HTF and α-His Western blotting revealed a single protein at the expected mass of 34 kDa (Sup-plementary Fig. 2B).

RNA-RNase III complexes were covalently crosslinked using UV-C in wild-type (untagged) and *rnc*-HTF tagged cultures grown in brain heart infusion (BHI) media to $OD_{578nm}$ 3.0. BHI was chosen as a rich medium to better facilitate capture of RNA–RNA duplexes associated with RNase III in exponential growing *S. aureus*. Our initial experiments (replicates 3–6) were prepared using a W5 small diameter UV-crosslinking unit (UVO3, UK) (1800 mJ UV-C, protocol A in Supplementary Methods) while replicates 1–2 were prepared using a Vari-X-linker (UVO3, UK) that allowed faster UV-crosslinking with lower doses (400 mJ UV-C, protocol B in Supplementary Methods)[27].

RNase III crosslinked RNA fragments were mapped to the JKD6009 transcriptome and had a good correlation between replicate experiments (Pearsons = 0.57–0.95) (Fig. 1c). The majority of reads (61.5–90%) mapped to rRNAs consistent with RNase III processing of this abundant species (Fig. 1d)[28,29]. Coding sequences (3–29%), 3′UTRs (1.5–3.8%), sRNAs (0.9–2.6%), and 5′UTRs (0.4–2.3%) were the next most abundantly recovered RNA classes (Fig. 1d). Plotting the cumulative read count across all CDS indicated that RNase III bound strongly to the 5′ end of mRNAs (Fig. 1e) and plotting RNase III binding relative to the start of 5′ UTRs and the end of 3′ UTRs indicated enrichment of RNase III within the UTRs of mRNAs (Fig. 1f). RNase III-bound 5′ and 3′ UTRs were generally longer and more structured compared with UTRs that did not bind RNase III, consistent with recognition of double-stranded RNA structures formed *in cis* within the UTR (Fig. 1g). These results are also consistent with analyses of RNase III in *S. pyogenes* where RNase III preferentially cleaves UTRs[30]. We additionally recovered RNase III interactions with the known substrates *cspA* and RNAIII, confirming that our analysis recovers bona fide targets (Supplementary Fig. 2C)[21,24]. We were not able to recover a statistically significant RNA sequence representing a binding motif within the RNase III read peaks, potentially reflecting broad read peaks recovered by RNase III UV-crosslinking and/or a significant role of in trans RNA–RNA interactions in forming the double-stranded RNA substrate for RNase III binding.

**RNase III-CLASH captures RNA–RNA interactions**. We have previously shown that our UV-crosslinking approach facilitates proximity-dependant ligation of RNA–RNA interactions[13]. We used the software package *hyb*[31] to extract hybrid reads that represent RNA–RNA interactions and filtered for interactions with an FDR < 0.05 (detailed in Supplementary Methods). We

collated our hybrid reads with additional RNase III-CLASH data generated in a parallel study by McKellar et al.[32] utilising TSB and RPMI-1640 media (Supplementary Data 3). We recovered 13,530 unique hybrid reads (21,680 in the collated dataset), representing 822 statistically significant unique RNA–RNA interactions (1,420 in the collated datasets), including 133 sRNA-mRNA interactions (Supplementary Data 3). Consistent with our earlier dataset[13] many interactions are recovered in a single experiment with 117 interactions recovered in multiple inde-pendent CLASH experiments. We recovered 7 individual hybrid reads mapping to the previously identified interaction between *spoVG* mRNA and the sRNA SprX[7]. RNA interactions were recovered for a broad range of RNA classes (Fig. 2a). The RNase III-CLASH dataset contains a high proportion of rRNA interac-tions that map to multiple sites in the transcriptome and were removed by our mapping pipeline (Supplementary Methods). These interactions potentially represent rRNA structure or background signal from these abundant RNAs. We recovered 543 statistically significant mRNA-mRNA interactions with 15 interactions recovered in multiple independent experiments (Fig. 2b) suggesting that many mRNAs may be able to exert regulatory functions in trans. RNA interactions were enriched at start codons, in line with canonical regulatory interactions that occlude the ribosomal binding site (RBS) (Fig. 2c). RNA–RNA interactions recovered by RNase III-CLASH had a significantly lower free energy than randomly shuffled RNA pairs, as did sRNA-target RNA interactions, consistent with hybrid reads representing in vivo RNA–RNA interactions (Fig. 2d).

The collated (BHI, TSB, and RPMI-1640) *S. aureus* sRNA interactome contains 287 nodes and 256 sRNA interactions (Fig. 3 and Supplementary Data 3). We independently verified a subset of sRNA-mRNA interactions within the RNase III-CLASH network using a two-plasmid system for high-level, constitutive expression of both sRNA and mRNA[7]. sRNA–mRNA interac-tions were chosen from the collated interaction network that had varying degree of hybrid counts. Using a SpoVG-GFP transla-tional fusion we demonstrated that SprX2 (the second copy of SprX that contains 6 SNPs and a single nucleotide deletion) is able to repress SpoVG expression consistent with earlier work[7]. Interestingly, the sRNA SprD is required for *S. aureus* pathogenicity[33] and we show it is also able to repress SpoVG expression through an imperfect 34-nt base-pair interaction at the 5′ UTR of the mRNA that likely occludes the RBS (Fig. 2e). The sRNA RsaA has previously been shown to repress MgrA expression and we have independently confirmed this interaction[21,34] (Fig. 2f). Point mutations within the predicted seed regions of either RsaA or MgrA significantly reduced repression, and repression could be partially restored when the complementary point mutations were expressed together, indi-cating a direct interaction between RsaA and MgrA in vivo (Fig. 2f). Many novel sRNA–mRNA interactions were positioned within mRNA coding sequences such as sRNA11-*agrA* and RNAIII-*murQ*. Using qRT-PCR to measure transcript abundance we confirmed that sRNA11 promotes *agrA* mRNA accumulation (Fig. 2g) and RNAIII destabilises *murQ* mRNA (Fig. 2h) indicating that these RNase III-CLASH interactions identified are functional.

These data indicate that RNase III-CLASH detects in vivo RNA–RNA interactions including functional sRNA-mRNA interactions providing a condition-specific snapshot of the *S. aureus* sRNA interactome.

**A novel regulatory RNA is required for intermediate vanco-mycin tolerance in VISA**. We next asked if sRNA interactions identified within our collated RNase III-CLASH network

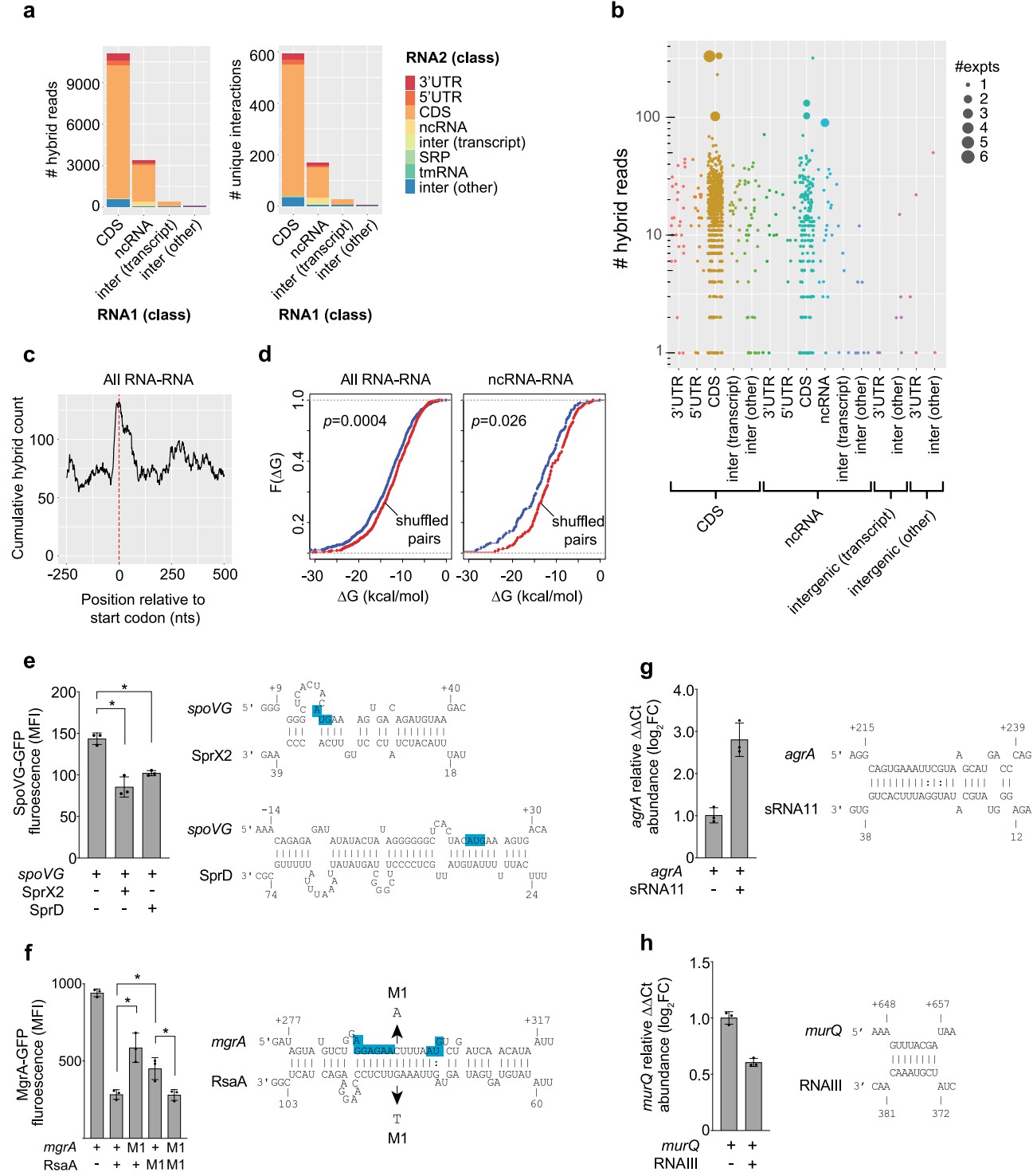

contributed to increased susceptibility to vancomycin in the VISA strain JKD6008[4]. Regulatory RNA interactions with mRNAs that are linked to the development of vancomycin tolerance[3], or up-regulated during vancomycin treatment[5] were selected for further analysis. CRISPR interference[35] (CRISPRi) was used to knock-down expression of six putative regulatory RNAs and the reduced expression was confirmed by Northern blot (Supplementary Fig. 3A). An agar spot dilution assay was used to assess vancomycin-sensitivity of each regulatory RNA knock-down in JKD6008. In the absence of vancomycin, the CRISPRi knock-downs had comparable growth to the vector-only control (Fig.

4ai). However in the presence of a sub-inhibitory concentration of vancomycin (3 μg/mL), growth of the regulatory RNA knockdown annotated as sRNA275 (here termed *vigR* 3′UTR) was reduced 1000-fold (Fig. 4aii). Broth microdilution indicated that the *vigR* 3′UTR CRISPRi knockdown was vancomycin sensitive with an MIC of 2–4 μg/mL (Supplementary Fig. 3B). In liquid MH, growth of the *vigR* 3′UTR CRISPRi knockdown was severely attenuated (1.91-fold increase in lag phase, $p = 0.0012$; 1.25-fold decrease in maximum $OD_{600nm}$, $p = 0.00054$) in the presence of vancomycin, similar to VSSA strain JKD6009 (Fig. 4b and Supplementary Fig. 3C). These data show that *vigR* 3′UTR

**Fig. 2 RNase III-CLASH recovers RNA–RNA interactions in *S. aureus* JKD6009. a** Histogram of the RNA classes recovered by RNase III-CLASH expressed as the number of hybrid reads (*left*) and number of unique RNA–RNA interactions (multiple hybrid reads can represent one RNA–RNA interaction) (*right*). **b** Distribution of the of hybrid reads representing each RNA–RNA interaction recovered by RNase III-CLASH. Each RNA–RNA interaction type is indicated below and the number of independent experiments containing the interaction is indicated by the size of the data point. **c** Cumulative count of RNA–RNA interactions at start codons (indicated by the red dashed line). **d** Cumulative distribution function of RNA–RNA interaction strength ($\Delta G$, kcal/mol) for all RNA–RNA interactions recovered (*left*), or ncRNA–RNA interactions (*right*). Pairs of interacting RNAs were randomly shuffled and the distribution of interaction strength of randomly paired RNAs is shown in red. A two-sided Kolmogorov–Smirnov test was used to calculate *p*-values. **e**-**h** RNA–RNA interactions recovered by RNase III-CLASH are functional. Constitutively transcribed GFP translational fusions to SpoVG (**e**) and MgrA (**f**) were expressed in *S. aureus* RN4220 with or without transcription of cognate sRNAs (indicated below) and median fluorescence intensity (MFI) measured using flow cytometry. Histogram heights represent mean MFI and error bars represent the standard deviation (SD) from $n = 3$ biological replicates. Significance was calculated using a two-sided *t*-test. *$p < 0.05$. Predicted interactions between RNAs recovered in hybrid reads are indicated (*right*). Blue boxes indicate the ribosomal binding site and start codon for each mRNA. **f** Compensatory point mutations (M1) were introduced into *mgrA* and RsaA (indicated by arrows). MFI was measured for combinations of *mgrA* and RsaA M1 mutants (indicated below histogram). For interactions within the CDS, qRT-PCR was used to quantify target mRNA abundance (relative to *gapA*) in *S. aureus* RN4220 constitutively transcribing the sRNAs sRNA11 (**g**) or RNAIII (**h**) from the vector pICS3 (indicated below plot). The mRNAs *agrA* (**g**) and *murQ* (**h**) are expressed from the chromosomal loci. Histogram heights represent mean relative abundance and error bars indicate standard error from $n = 3$ biological replicates. *p*-values were calculated using a two-sided *t*-test. Predicted interactions between RNAs recovered in hybrid reads are indicated (*right*).

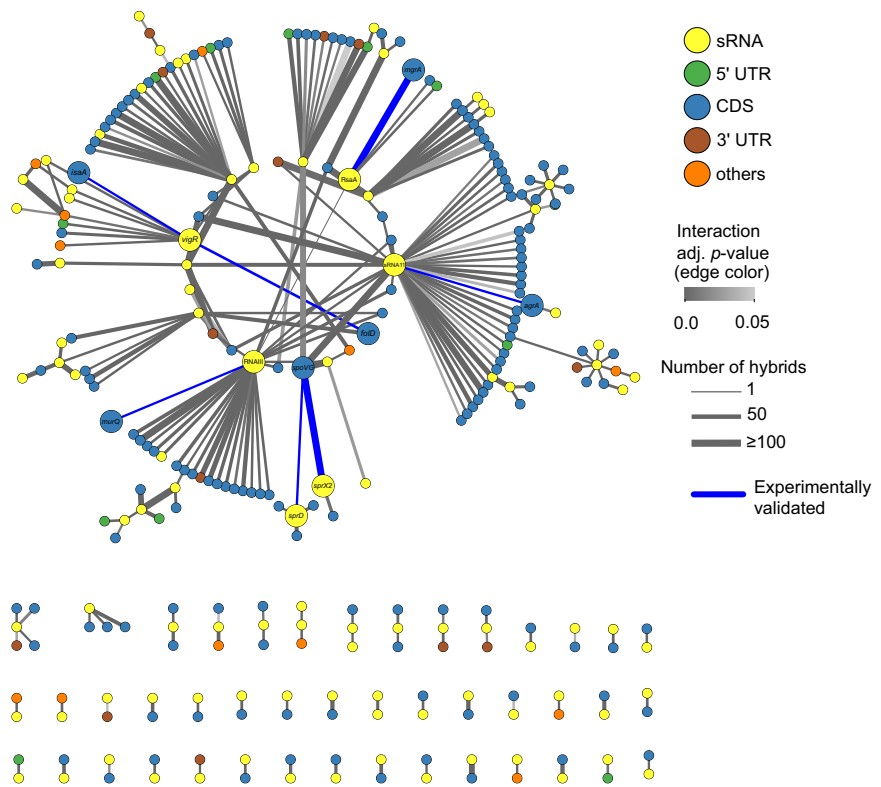

**Fig. 3 Small RNA interactome of *S. aureus* JKD6009.** Individual RNAs are indicated as nodes (circles) and coloured according to RNA class (*top right*). RNA–RNA interactions are indicated as edges (lines) and coloured according to FDR adjusted probability (*middle right*). The discrete probability P(X = n) of recovering each hybrid was modelled using a binomial distribution and combined between replicate experiments using the Fishers method. A detailed description of the statistical analysis performed on the RNase III-CLASH dataset is presented in Supplementary Methods. The line thickness of RNA–RNA interaction edges are weighted to indicate the number of hybrid reads captured for each interaction (*bottom right*). RNA–RNA interactions that are experimentally validated in the text are labelled at the node and edges are coloured blue.

(previously annotated as sRNA275) is required for vancomycin intermediate resistance in JKD6008, and that knocking-down expression of *vigR* 3′UTR reverts the strain to a vancomycin-sensitive phenotype.

**vigR is a regulatory mRNA that controls a glycopeptide-specific intermediate tolerance**. Term-seq identified a transcription termination site at the 3′ end of the long *vigR* 3′UTR and our dRNA-seq data indicated a primary TSS upstream of the uncharacterised YtxH-domain protein, *E0E12_RS09390* (*vigR*) (Fig. 4c). We did

not identify additional TSSs, processing sites, or termination sites within *E0E12_RS09390* or the UTR boundaries. Northern blot analysis probing for *vigR* 3′UTR identifed a long transcript (~1.1 kb) indicating that sRNA275 is the 3′UTR of the *E0E12_RS09390* mRNA (Fig. 4d). Based on these results and later observations we have named this mRNA the vancomycin-intermediate and glycopeptide resistance mRNA (*vigR*). Northern blot analysis indicated that *vigR* expression is up-regulated in the VISA strain JKD6008 compared to JKD6009, consistent with a role in intermediate vancomycin tolerance (Fig. 4d and Supplementary Fig. 3D). Northern blot analysis also indicated that *vigR*

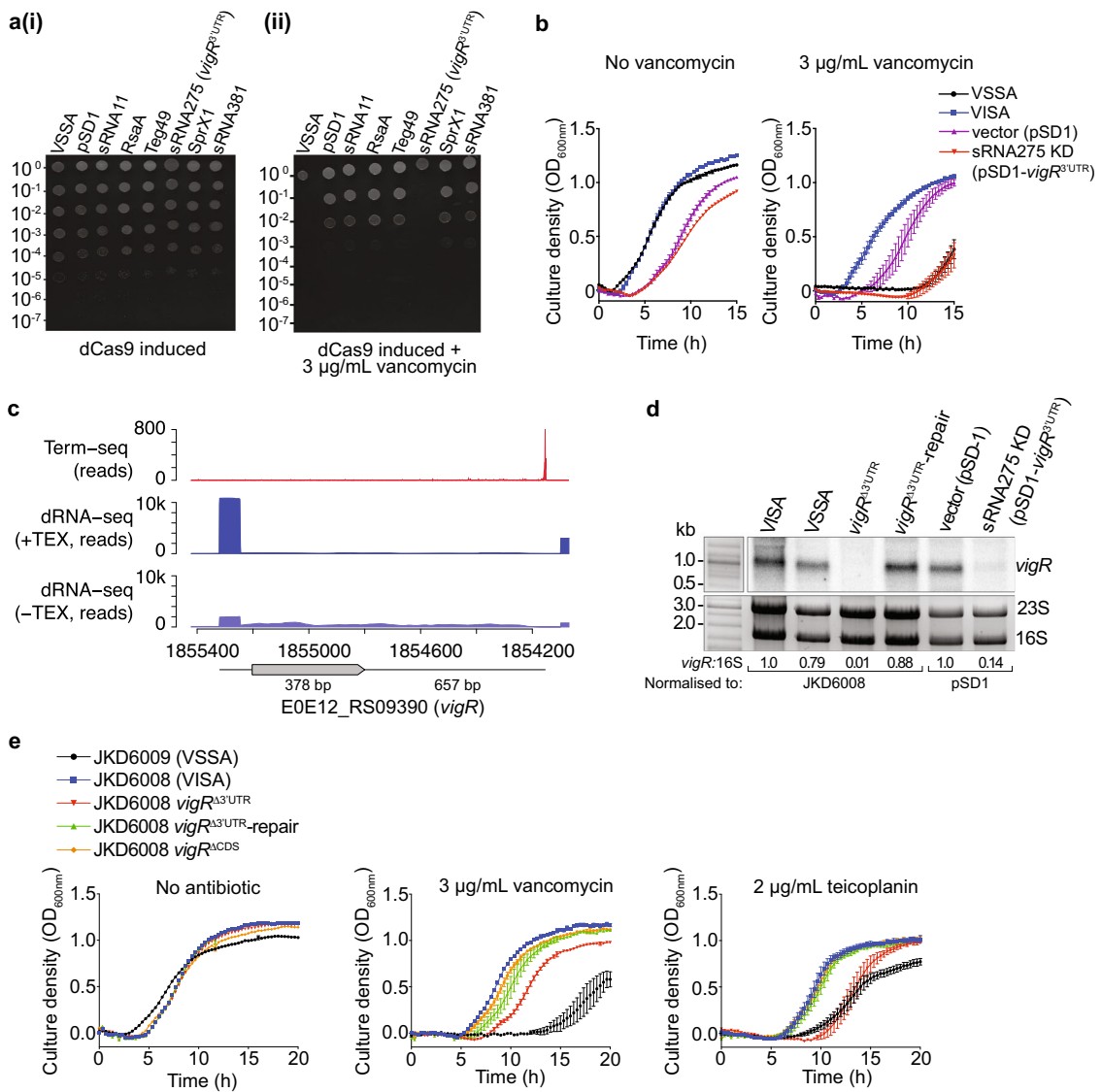

**Fig. 4 The regulatory RNA, sRNA275 (*vigR* 3′UTR), is required for intermediate-vancomycin tolerance in *S. aureus* JKD6008. a** Spotted dilution assays quantified vancomycin tolerance in sRNA induced CRISPRi knockdown strains. Small RNAs targeted by CRISPRi sgRNAs are indicated above. Culture dilution is indicated left. The cultures were grown in the absence (i) or presence (ii) of 3 µg/mL vancomycin. **b** Growth curves of VISA (JKD6008), VSSA (JKD6009), pSD1 (vector only control, JKD6008 pSD1), and pSD1-*vigR*³′UTR (JKD6008 *vigR* 3′UTR CRISPRi knockdown). Cultures were grown in MH media with or without sub-inhibitory vancomycin (3 µg/ml). **c** Mapping *vigR* mRNA 5′ and 3′ ends. Plots indicate raw read data mapping to *vigR* (E0E12_RS09390) generated using Term-seq (*top*), or dRNA-seq with (*middle*) or without (*bottom*) terminator exonuclease (TEX). The position of the E0E12_RS09390 CDS is indicated below. **d** Northern blot analysis of *vigR* mRNA. Total RNA was extracted from strains indicated (*top*) and probed for *vigR* RNA. Excepting VSSA (JKD6009), all strains are derived from JKD6008. Sybr Green II stained 23S and 16S rRNAs are indicated below as a loading control. Quantification of the ratio of 16S rRNA to *vigR* by densitometry is indicated below. **e** Growth curves for VSSA (JKD6009), VISA (JKD6008), JKD6008 *vigR*^Δ3′UTR, JKD6008 *vigR*^Δ3′UTR-repair, and JKD6008 *vigR*^ΔCDS mutants. Strains were grown in MH media supplemented with antibiotics indicated above.

transcript levels are increased between mid-log and stationary phase during growth in BHI (Supplementary Fig. 3D), but appear consistent throughout the growth phases in RPMI-1640 media (Supplementary Fig. 3E). We also observed a single transcript band (~1.1 kb) in RPMI-1640 media, confirming no processing events or independent transcription of the *vigR* 3′UTR in this infection-relevant condition (Supplementary Fig. 3E).

Our results suggested that the *vigR* 3′UTR may confer vancomycin tolerance through *cis* regulation of the VigR protein. To determine the relative contribution of each region to intermediate-vancomycin tolerance, clean deletions of both the 3′UTR (*vigR*^Δ3′UTR) and CDS (*vigR*^ΔCDS), and a chromosomally

repaired *vigR*^Δ3′UTR (*vigR*^Δ3′UTR-repair, restoring the wild type genotype) were constructed in JKD6008 (schematic representation of constructs in Supplementary Fig. 3F). These strains were confirmed using Northern blot analysis (Fig. 4d), qRT-PCR (Supplementary Fig. 3G) and whole genome sequencing. We find that the *vigR* 3′UTR is required for *vigR* CDS stability (CDS transcript levels are 37.5% c.f. WT, Supplementary Fig. 3Gi). The 3′UTR of *vigR* is more stable in the absence of the CDS (68.2% c.f. WT; Supplementary Fig. 3Gii).

The *vigR*^ΔCDS strain had a slight growth defect in liquid MH and grew to similar levels in the presence of 3 µg/mL vancomycin (Fig. 4e and Supplementary Fig. 4A). In contrast, the *vigR*^Δ3′UTR

deletion grew similar to the parent VISA strain in liquid MH but was sensitive to vancomycin and this could be partially restored by repairing the $vigR^{\Delta 3'UTR}$ deletion ($vigR^{\Delta 3'UTR}$-repair, Fig. 4e and Supplementary Fig. 4A). Deletion of the $vigR$ 3′UTR also sensitised the VSSA strain JKD6009 to vancomycin (Supplementary Fig. 4B), indicating that $vigR$ also contributes to vancomycin tolerance in the MRSA background, albeit at a lower MIC.

We next assessed if the $vigR$ 3′UTR deletions had altered susceptibility to other antibiotics used to treat *S. aureus* infections. The $vigR$ 3′UTR deletion, but not the CDS deletion, was sensitive to an intermediate level of the last-line antibiotic teicoplanin (2 μg/mL) in both the VISA (Fig. 4e and Supplementary Fig. 4A) and VSSA backgrounds (Supplementary Fig. 4B), and was fully restored in the repaired strains. Neither $vigR$ 3′UTR deletions were significantly more sensitive to fosfomycin, tigecycline, or ampicillin than their respective parents strains (Supplementary Fig. 4B, C). Collectively, these results indicate that the $vigR$ 3′UTR controls a glycopeptide-specific response, independent of the VigR protein.

### $vigR$ 3′UTR regulates the expression of cell wall metabolism genes.

Our RNase III-CLASH analysis indicated that $vigR$ 3′UTR acts as a regulatory hub (Fig. 5a). To understand how the expression of $vigR$ target RNAs were controlled and to gain further insight into the mechanism of intermediate glycopeptide tolerance, we used RNA-seq to measure RNA abundance in both the $vigR^{\Delta 3'UTR}$ and $vigR$ 3′UTR CRISPRi knockdown. These analyses identified 117 transcripts including 16 sRNAs that were differentially expressed in both the $vigR$ 3′UTR deletion and knockdown (FDR ≤ 0.05). The addition of sub-inhibitory vancomycin (2 μg/mL) for 10 min did not reveal additional transcriptional changes (Supplementary Fig. 5). Ontological clustering of differentially expressed transcripts indicated that terms associated with 'carbohydrate transport and metabolism', 'amino acid transport and metabolism', and 'cell wall, envelope, and membrane biogenesis' were enriched (Fig. 5b). Differentially expressed transcripts involved in cell wall and envelope biosynthesis included the downregulation of $dat$[36], $lytM$[37], the $dltXABCD$[38] operons and the lytic transglycosylases $isaA$ and $sceD$, along with the upregulation of $murAGQ$[39] (Fig. 5c). Notably, $isaA$, $sceD$, and $lytM$ also belong to the WalKR regulon[40] suggesting that the $vigR$ mRNA and WalKR regulons at least partly overlap. There was no change in transcript abundance of $walKR$ or $walHI$ in the $vigR^{\Delta 3'UTR}$ or knockdown strain, indicating that cross-regulation does not occur at the transcriptional level.

The $vigR$ CLASH targets $folD$ and $isaA$ were reduced in the $vigR^{\Delta 3'UTR}$ and/or $vigR$ knockdown strain indicating that $vigR$ mRNA–mRNA interactions stabilise these transcripts (Fig. 5a, c). The $folD$ mRNA (reduced in the knockdown strain, FDR = 0.0047, $log_2$FC = −1.21), involved in folate metabolism, produces tetrahydrofolate which forms as a key metabolite for amino acid (histidine) and nucleotide (purine) biosynthesis. The $vigR$ and $folD$ mRNAs are predicted to form an extensive 145-nt RNA–RNA duplex including 63-nts of perfect Watson–Crick base-pairing (Fig. 5d). We used EMSA to confirm that the $vigR$ 3′UTR stabilises $folD$ mRNA through a direct interaction. Titrating the $vigR$ 3′UTR with either a $^{32}$P-labelled full-length $folD$ mRNA or a $^{32}$P-labelled $folD$ sub-fragment incorporating the 145-nt duplex site shifted the $^{32}$P-labelled $folD$ to a slower migrating species consistent with the formation of a $vigR$-$folD$ RNA duplex (Fig. 5e).

Collectively, these results indicate that $vigR$ mRNA has profound effects on the abundance of transcripts required for cell wall metabolism, and confirm the novel interaction with $folD$ mRNA identified by RNase III-CLASH that increases the abundance of this transcript.

### $isaA$ mRNA is directly regulated by the $vigR$ 3′UTR.

The abundance of $isaA$ was reduced in the $vigR$ 3′UTR deletion (FDR = 0.00012, $log_2$FC = −1.02) and knockdown strain (FDR = 0.033, $log_2$FC = −0.69) (Fig. 5c), and we confirm this result using Northern blot analysis (Fig. 6a). To confirm $vigR$ 3−UTR interacts with $isaA$, we employed a two-plasmid system to constitutively transcribe $vigR$ 3′UTR and an $isaA$-GFP translational fusion[7]. This construct does not include the $isaA$ promoter and high-level, constitutive transcription of $isaA$-GFP from $P_{tufA}$ uncouples expression from native transcriptional regulation. In the presence of the $vigR$ 3′UTR, translation of $isaA$-GFP was increased 58% ($p = 0.0001$), confirming that the $vigR$ 3′UTR promotes $isaA$ mRNA expression independently of transcriptional regulation (Fig. 6b). EMSA was used to verify a direct interaction between $vigR$ 3′UTR and $isaA$ mRNA in vitro. A slower migrating $vigR$-$isaA$ duplex was formed when titrating $^{32}$P-labelled $isaA$ or $vigR$ 3′UTR against each other (Fig. 6c, d). To determine the interaction site, we titrated radiolabelled $vigR$ 3′ UTR against sub-fragments of $isaA$ (Fig. 6e). These analyses indicated that the $vigR$ 3′UTR interacts with the 3′ region of the $isaA$ coding sequence ($isaA$ frag-C, Fig. 6f), consistent with the position of the RNA–RNA hybrid captured by RNase III-CLASH. A fainter $vigR$-$isaA$ frag-A complex was also formed (Supplementary Fig. 6A). Labelled $vigR$ 3′UTR was not able to shift $isaA$ frag-B (Supplementary Fig. 6B) indicating that the $isaA$ fragment C interaction is specific. IntaRNA analysis of $vigR$ 3′UTR-$isaA$ frag-C identified an imperfect 86-nt interaction duplex (50-nt of complementarity) that included the GC-rich RNase III-CLASH hybrid read (Fig. 6g). To further resolve the interaction site, competitor oligonucleotides antisense to either $vigR$ 3′UTR (C1–C4) or $isaA$ frag-C (C5–C8) tiled across the duplex site were added into the EMSA reaction (500× excess concentration). This analysis revealed that competitor oligonucleotides at the C1 and C5 site antisense to $vigR$ and $isaA$, respectively, were able to effectively compete away radiolabelled $vigR$ 3′UTR-$isaA$ frag-C duplex formation (Fig. 6h). Additionally, the competitor oligonucleotides at the C2 site antisense to $vigR$ 3′UTR was also able to effectively compete away radiolabelled $vigR$ 3′UTR, indicating that the 3′ end of the 3′UTR duplex (position + 1069-1038) is crucial in vitro for $vigR$ 3′UTR-$isaA$ frag-C duplex formation. To complement the EMSA, we titrated $isaA$ frag-C with $vigR$ 3′UTR in the presence of RNase T1 and ShortCut RNase III followed by $^{32}$P-labelled-primer extension (Supplementary Fig. 6C). In agreement with the competitor oligonucleotides at the C1 and C5 site, binding of $isaA$ frag-C induced ShortCut RNase III cleavage most notably at position C + 1063 (Fig. 6g and Supplementary Fig. 6D).

To understand the role of RNase III in the regulation of $isaA$, a deletion of $rnc$ was constructed in JKD6009. Using qRT-PCR, we confirm the increased abundance of the $isaA$ transcript in the Δ$rnc$ strain (Fig. 6i). qRT-PCR showed a statistically significant increase of 35.5% (± 3.9, $p = 0.00046$; Fig. 6i) relative to WT, consistent with RNase III processing of $isaA$.

These results indicate that the $vigR$ 3′UTR is able to posttranscriptionally promote $isaA$ expression in vivo and directly interacts in vitro with the 3′ end of $isaA$ identified by RNase III-CLASH.

### $vigR$ 3′UTR regulation of $isaA$ impacts cell wall thickness and glycopeptide-intermediate tolerance.

To determine the impact of $isaA$ regulation on intermediate vancomycin resistance, an $isaA$ deletion and CRISPRi knockdown strain was constructed in VISA strain JKD6008. IsaA has previously been shown to confer salt resistance and we independently verified this phenotype for our Δ$isaA$ strain in both BHI and MH media[41] (Supplementary

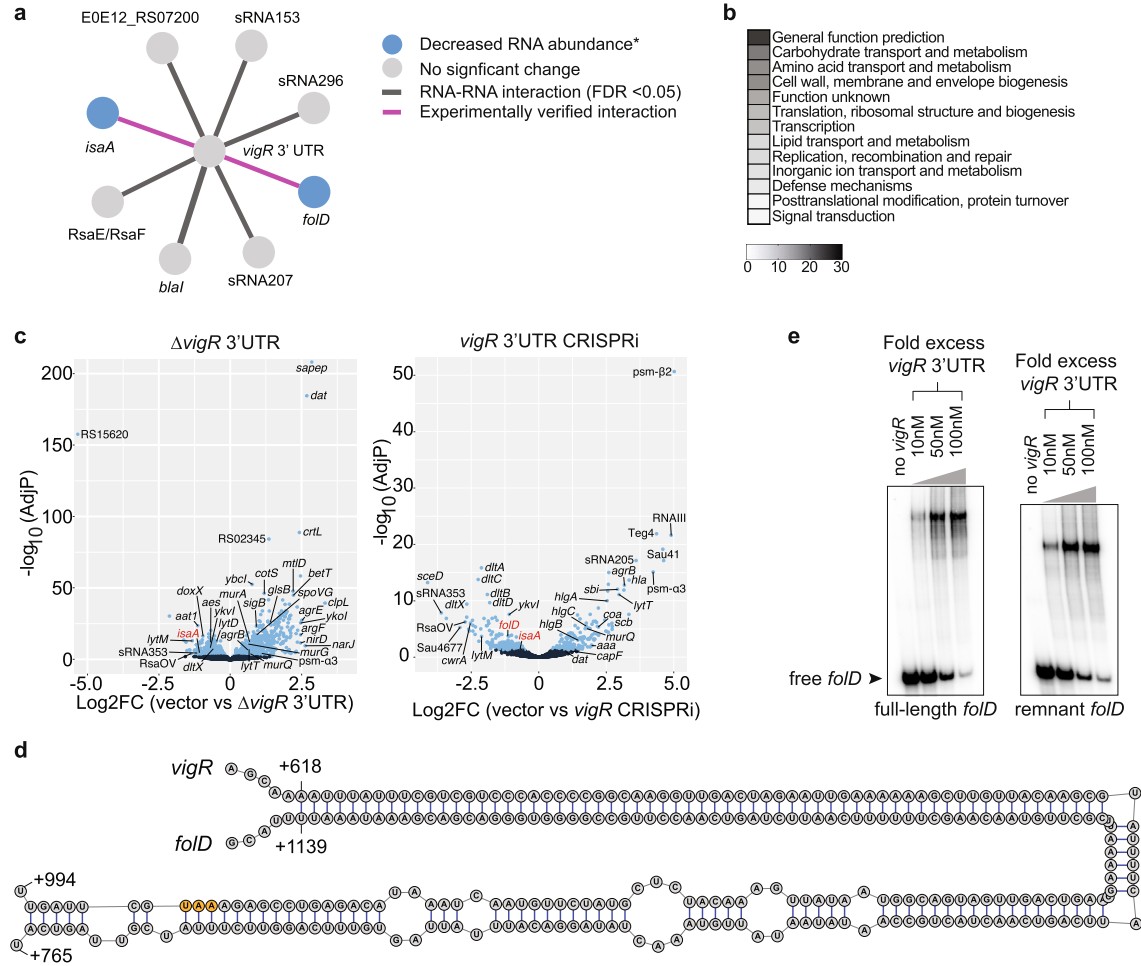

**Fig. 5 The 3′ UTR of *vigR* regulates cell wall metabolism. a** Sub-network of RNase III-CLASH RNA–RNA interactions with *vigR* 3′UTR. *Target RNA abundance significantly decreased in JKD6008 *vigR*^Δ3′UTR or CRISPRi knockdown strains (see panel **c**). **b** Clusters of orthologous (COG) classes detailed for differentially expressed transcripts found in both *vigR*^Δ3′UTR and CRISPRi knockdown strains (adjusted *p* < 0.05). The most abundant COG classes are listed in numerical order. **c** Volcano plots of differential expression in JKD6008 vs JKD6008 *vigR*^Δ3′UTR (*left*) and JKD6008 pSD1 (vector control) vs pSD1::sgRNA-*vigR* 3′UTR (CRISPRi knockdown) (*right*). Statistically significant changes in RNA abundance are indicated in light blue (adjusted *p* < 0.05). DESeq2 calculates a two-sided *p*-value based on the Wald statistic that is corrected for multiple testing using the Benjamini–Hochberg method. Transcripts for *isaA* and *folD* are indicated by red text. **d** Base pairing interaction predicted to form between the *vigR* 3′UTR and *folD* mRNA. The *folD* stop codon is indicated in orange. **e** EMSA analysis of RNA–RNA interactions between radiolabelled full-length *folD* mRNA and the *vigR* 3′UTR (*left*), and radiolabelled sub-fragment of *folD* (*right*). Approximately 50 fM of *folD* was radiolabelled and titrated against increasing concentrations of *vigR* 3′UTR (indicated top). The migration of free *folD* mRNA is indicated by the black arrow.

Fig. 7A). A common feature of clinical VISA isolates is a thicker cell wall[3]. We used TEM to quantify the cell wall thickness of JKD6009 (VSSA), JKD6008 (VISA), *vigR*^Δ3′UTR, Δ*isaA*, and JKD6008 pSD1 and pSD1-*isaA* knockdown (Fig. 6j). We confirmed the increased cell wall thickness reported for VISA isolates (25.84 c.f. 22.77 nm; Fig. 6j and Supplementary Fig. 7B).

Interestingly cell wall thickness measurements of the *vigR*^Δ3′UTR strain revealed a decrease in cell wall thickness to 24.13 nm (*p* = 0.058) when compared to the isogenic VISA parent strain, suggesting that *vigR* 3′UTR influences cell wall architecture in *S. aureus* (Supplementary Fig. 7C, D).

Most notably, both the *isaA* deletion and knockdown strain had a significantly reduced cell wall thickness when compared to their isogenic parent strains (Fig. 6j and Supplementary Fig. 7B). Deletion of *isaA* also sensitised JKD6008 to vancomycin but did not completely recapitulate the acute vancomycin sensitivity of the *vigR*^Δ3′UTR strain indicating that additional regulatory effects contribute to vancomycin sensitivity (Supplementary Fig. 7E). However, the Δ*isaA* strain was sensitive to teicoplanin indicating

that activation of *isaA* is likely responsible for teicoplanin sensitivity in the *vigR*^Δ3′UTR strain (Supplementary Fig. 7E).

These results demonstrate that *isaA* contributes to cell wall thickening that is partly responsible for the intermediate vancomycin resistance of VISA isolate JKD6008.

## Discussion

Methicillin-resistant *Staphylococcus aureus* has become increasingly common in both community and healthcare settings. Treatment of MRSA infections is limited to last-line antibiotics and vancomycin is the drug of choice for severe MRSA septicaemia. Intermediate vancomycin resistance is the most common cause of vancomycin treatment failure and arises through a heterogenous collection of point mutations that often lead to reduced autolysis and cell wall thickening which reduces vancomycin permeability[3]. Previous transcriptome profiling experiments suggested that regulatory RNA responses may play critical roles in resistance to last-line antibiotics[5]. Here we have used RNase III-CLASH to identify an mRNA that is required for

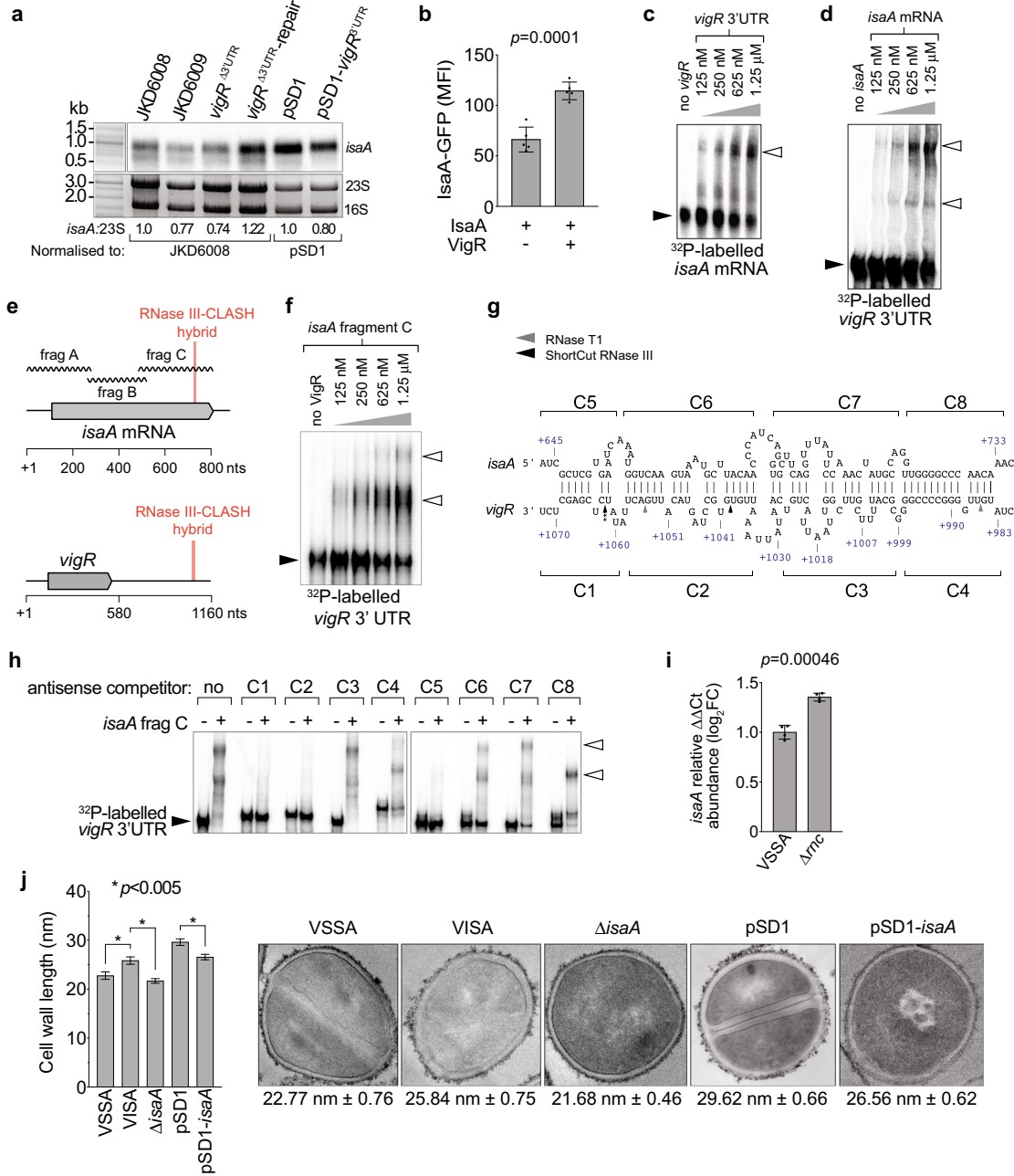

intermediate vancomycin resistance and functions as a regulatory RNA 'hub' in our network. CRISPRi knockdown of sRNA275 (here termed *vigR* 3′UTR) reverted the clinical VISA isolate JKD6008 to vancomycin-sensitive. Northern analysis, dRNA-seq, and Term-seq mapping of RNA 5′ and 3′ ends demonstrated that sRNA275 is the 3′UTR of the upstream coding sequence *E0E12_RS09390* (SAA6008_01724) encoding a hypothetical YtxH-domain protein (here termed VigR). The *vigR* 3′UTR is unusually long for a prokaryotic 3′UTR at 657-nt (median 40-nt *Bacillus subtilis*, 75-nt in this study, and 88-nt in *Methanosarcina mazei*)[42], and is substantially longer than the 378-nt CDS encoded within the *vigR* mRNA. We demonstrate that the 3′UTR, but not the CDS is required for intermediate vancomycin tolerance indicating that the mRNA, and not the protein mediates these regulatory effects. Bacterial 3′UTRs can have both *cis*-[43–45], and *trans*-acting regulatory activity. Examples of *trans*-acting regulatory sRNAs encoded within 3′UTRs have recently been described[46,47]. These 3′UTR sRNAs can be transcribed as independent transcripts or released from the mRNA by RNase cleavage, and function independently of the parent mRNA[48]. Here we find that the *vigR* 3′UTR is neither independently transcribed, nor processed from the *vigR* mRNA indicating that it is a regulatory mRNA. To our knowledge, *vigR* is the fourth example of a bacterial mRNA that has *trans*-acting regulatory functions. In *S. aureus*, the CDS region of *gdpS* was shown to directly interact and stabilise the 5′UTR of *sarS*, however, the mechanism was not determined[49]. In the gram-positive pathogen, *Listeria monocytogenes*, interactions between the 3′UTR of *hly* and the 5′UTR of *prsA* block exonucleolytic digestion of *prsA* mRNA by RNase J1[50]. In *Streptococcus mutans*, the 5′UTR of *irvA* interacts with the CDS of *gbpC* where it occludes an RNase J2 endonucleolytic cleavage site[51]. In both cases, the mRNA-mRNA interaction stabilise the target mRNA by protecting the transcript from ribonuclease processing or degradation. In a slight variation on this Gram-positive theme, we show that the *vigR* 3′UTR interaction with the CDS of *folD* and *isaA* mRNAs

**Fig. 6 The lytic transglycosylase IsaA is regulated by the 3′UTR of *vigR*. a** Northern analysis of *isaA* abundance in VISA (JKD6008) and isogenic *vigR*$^{\Delta 3'UTR}$, *vigR*$^{\Delta 3'UTR}$-repair, along with JKD6008 pSD1 and pSD1-*vigR* 3′UTR. Sybr Green II stained 23S and 16S rRNA are shown as loading controls. Quantification of the ratio of 23S rRNA to *isaA* by densitometry is indicated below. **b** Quantification of expression of a constitutively transcribed IsaA-GFP fusion (pCN33::*isaA-gfp*) with or without expression of *vigR* 3′UTR (pICS3::*vigR*) (indicated below). Median fluorescence intensity (MFI) is reported on the *y*-axis. Histogram heights represents the mean MFI and error bars indicate standard error from $n = 5$ biological replicates. The *p*-value is calculated using a two-sided *t*-test. **c** EMSA analysis of interactions between *vigR* 3′UTR and full-length *isaA* mRNA. 50 fM of radiolabelled *isaA* mRNA was titrated against increasing concentrations of *vigR* 3′UTR (*top*). **d** 50 fmol of radiolabelled *vigR* 3′UTR was titrated against increasing concentrations of full-length *isaA* mRNA. **e** Schematic of the RNase III-CLASH hybrid read within *isaA* mRNA and *vigR* mRNA. The *isaA* mRNA was synthesised as sub-fragments (frag A–C, wavy lines) ~300-nt in length to be used for EMSA. **f** *isaA* fragment C (frag-C). Concentrations of *isaA* are indicated (*top*). Black arrowheads indicated migration of free, radiolabelled RNA and open arrowheads indicate slow migrating *vigR-isaA* duplexes. **g** Predicted interaction between RNAs within the *isaA-vigR* hybrid read. The start and end positions of the RNA–RNA duplex are indicated respresentative of the mRNA TSS. The antisense oligonucleotide competitors used for EMSAs are detailed above and below their respective mRNAs (C1–C83). The cleavage sites of RNase T1 and ShortCut RNase III detected by primer extension are indicated by the grey and black arrows, respectively (also see Supplementary Fig. 6C, D). **h** EMSA analysis of interactions between *vigR* 3′UTR and *isaA* frag-C (0 or 1.25 μM concentration). Black arrowheads indicate migration of free radiolabelled *vigR* 3′UTR RNA. Antisense competitor oligonucleotides C1–C8 (indicated on top) were spiked in at 500× excess concentration. Open arrowhead indicates slow migrating *vigR-isaA* frag-C duplexes. **I** Quantitative RT-PCR for *isaA* mRNA abundance (relative to *gapA*) in VSSA and Δ*rnc* strain grown in BHI to an OD 578$_{nm}$ of 3.0. Histogram height represents the mean and error bars indicate the standard deviation (SD) from $n = 3$ biological replicates. The *p*-value is calculated using a two-sided *t*-test. **J** Histogram of cell wall thickness for VSSA, VISA, and *isaA* deletion and knockdown strains (*left*). Error bars represent standard error of the mean (SEM) from $n = 100$ measurements. The *p*-value is calculated using a two-sided *t*-test. *$p < 0.005$. (*right*) Representative transmission electron microscopy (TEM) images of VSSA (JKD6009), VISA (JKD6008), and JKD6008 derivatives. The average cell wall thickness and standard error are shown below.

also stabilise these transcripts likely by blocking RNase III endonucleolytic processing within these mRNAs.

Deletion of the *vigR* 3′UTR led to a modest reduction in cell wall thickness and notable changes in the transcriptome. RNA-seq analysis of the *vigR*$^{\Delta 3'UTR}$ and knockdown strain indicated differential expression of 117 transcripts with many that clustered within the ontological term 'cell wall, membrane, and envelope biogenesis'. The addition of sub-inhibitory concentrations of vancomycin did not significantly alter the mutant transcriptome and *vigR* was not up-regulated by vancomycin, indicating that it promotes an intrinsically resistant physiology rather than an antibiotic-dependant response. This is supported by upregulated of *vigR* expression levels in the VISA strain JKD6008 when compared to the VSSA strain JKD6009. The transcriptional response to deletion or knockdown of the *vigR* 3′UTR was quite broad and we cannot rule out that interactions beyond those recovered by RNase III-CLASH and confirmed in vitro, may be due to secondary effects on transcription factors or general stress responses. We note that the lytic transglycosylases *isaA* and *sceD*, and autolysin *lytM* are positively regulated by the two-component sensor/kinase WalKR[40] and additional, indirect regulation of WalKR may be occurring. Indeed for *isaA*, indirect activation of WalKR and direct post-transcriptional activation of *isaA* mRNA though *vigR* would represent a feed-forward loop that is an increasingly common motif in regulatory RNA circuits[52].

Our methicillin-resistant *S. aureus* RNase III-CLASH dataset contains a surprisingly large proportion of interactions between coding sequences. Some of these may represent interactions with sRNAs that are embedded within CDSs and were missed by our transcriptome annotation[53,54]. However, given the abundance of CDS-CDS interactions, these results suggest that regulatory mRNA interactions, like *vigR-isaA* and *vigR-folD*, may be a much more widely utilised mechanism for coordinating gene expression in *S. aureus*. This may be due to the presence of 5′-3′ exoribonuclease activity (RNase J1 and J2) found in Gram-positive *Firmicutes*, which readily degrades free 3′UTR intermediates and may represent an evolutionary barrier to the evolution of 3′UTR-derived sRNA–mRNA interactions in *S. aureus* (discussed in ref. [55]). Regulatory mRNA 3′UTRs like *vigR* that are protected from processing at the 5′ end is likely a more widespread regulatory mechanism in *S. aureus* than previously appreciated.

Peptide-conjugated antisense oligonucleotides (ASOs) have gained renewed interest in recent years and allow specific targetting of RNAs for repression[56,57]. ASOs targeting regulatory RNAs like *vigR* may represent an effective therapeutic approach to re-sensitise VISA isolates to last-line vancomycin treatment.

## Methods

**Bacterial strains, plasmids, and culture conditions**. The bacterial strains, plasmids, and oligonucleotides used in this study are listed in Supplementary Data 4. *S. aureus* RN4220 and the JKD6009/JKD6008 (VSSA/VISA) pair strains were routinely cultured at 37 °C on solid or in liquid brain heart infusion (BHI, Merck) or Mueller–Hinton (MH, Merck) media. Antibiotics were routinely used in this study to select for plasmids in *S. aureus* at 10 μg/mL chloramphenicol and/or 10 μg/mL erythromycin, unless otherwise specified. *E. coli* DH5α and IM08B strains were routinely cultured at 37 °C on solid or in liquid Luria-Bertani (LB) media. Antibiotics were routinely used to select for plasmids in *E. coli* at 100 μg/mL ampicillin and 15 μg/mL chloramphenicol, unless otherwise specified. All bacterial strains were stored at −80 °C as stationary phase cultures with 16% (v/v) glycerol.

**Strain modifications**. *S. aureus* deletions for *vigR*$^{CDS}$, *vigR*$^{3'UTR}$, and *isaA* and chromosomal repair for *vigR*$^{3'UTR}$ were constructed using the pIMAY-Z vector and allelic exchange[58]. At least 500-nt flanking regions using primer pairs detailed in Supplementary Data 4 were amplified from the respective JKD6008 or JKD6009 gDNA using Phusion Hot Start Flex Polymerase (NEB). All amplified flanking regions were annealed together using splicing by overlap extension (SOE) PCR[58,59]. Mutants were passaged and selected on solid BHI and confirmed using allele-specific PCR (Supplementary Data 4). Loss of the pIMAY-Z vector was confirmed by 15 μg/mL chloramphenicol sensitivity and plasmid-specific PCR (Supplementary Data 4).

CRISPR interference (CRISPRi) transcriptional knockdown constructs in *S. aureus* were constructed using the pSD1 vector system[35]. Knockdown primer pairs (Supplementary Data 4) were annealed by heating to 94 °C for 2 min and then cooling 1.5 °C per min for 1 h. The annealed oligonucleotides were cloned into pSD1 at the SapI site using 10 U of T4 DNA ligase (Thermo) and transformed into chemically-competent *E. coli* DH5α. Constructs were confirmed by Sanger sequencing, transformed into electrocompetent *E. coli* IM08B, and then transformed into electrocompetent *S. aureus* JKD6008.

**Differential RNA-Seq (dRNA-Seq)**. An overnight stationary culture of *S. aureus* JKD6009 was diluted 1/100 and grown in liquid MH at 37 °C with 200 rpm shaking to an OD$_{600nm}$ 0.6. vancomycin (2 μg/mL) was added to antibiotic-treated cultures and allowed to grow for a further 10 min. Media-based cultures were additionally grown for a further 10 min at 37 °C to reflect the treated samples. Growth was halted by the addition of RNAProtect bacterial reagent (Qiagen) and incubated on ice for 10 min. Cells were harvested by centrifugation (3200 × *g* for 10 min) and underwent GTC-phenol:chloroform RNA extraction procedures[60]. cDNA libraries were prepared by Vertis Biotechnologie (Freising, Germany) as described previously[61] and sequenced on an Illumina NextSeq500 platform (75-cycle single-end reads) (Vertis Biotechnologie, Freising, Germany).

**Term-Seq**. *S. aureus* JKD6009 was prepared for total RNA extraction as described above, with the exception of using cold transcriptional stop solution (1:9

phenol:ethanol) to halt culture growth. Total RNA was DNase-treated using 10 U of RQ1 RNase-free DNase (Promega) at 37 °C for 30 min and ethanol precipitated. cDNA libraries were prepared by Vertis Biotechnologie (Freising, Germany) as described previously[15] and sequenced on an Illumina NextSeq500 platform (75-cycle single-end reads) (Vertis Biotechnologie, Freising, Germany). Detailed protocols for the analysis of the Term-seq data are presented in Supplementary Methods.

**Annotation of *S. aureus* JKD6009.** The ANNOgesic pipeline[16] was used to integrate transcriptomics data generated from RNA-seq, dRNA-seq[61], and Term-seq[15] to provide a detailed annotation of the *S. aureus* JKD6009 genome. In brief, the transcription start sites (TSS) were identified using the TSSPredator[62] module within the ANNOgesic pipeline. Rho-independent transcription termination sites were identified from the Term-seq data using custom R scripts and the 'peakPick' R library[63]. The ANNOgesic pipeline was used to analyse the transcriptomics data and the analyses resulted in a GFF file containing detailed annotations of *S. aureus* JKD6009 transcriptome. This GFF file was used in subsequent CLASH data analyses. The scripts from TermPick pipeline for the analysis of Term-seq data are available in the following GitHub repository (https://github.com/IgnatiusPang/TermPick). Detailed protocols for the annotation of the JKD6009 genome using ANNOgesic are presented in Supplementary Methods.

**RNase III-CLASH.** The chromosomal copy of RNase III (*rnc*) in *S. aureus* JKD6009 was tagged with the dual affinity tag, His6-TEV-FLAG (HTF). RNase III-CLASH was performed as described previously[13] using the JKD6009 WT and *rnc*.HTF strains with a total of six replicates. For a detailed description of RNase III-CLASH performed on initial replicates 3–6 (protocol A) and then replicates 1–2 (protocol B) and modifications to Waters et al. refer to Supplementary Methods. Briefly, JKD6009 and *rnc*.HTF strains were used to inoculate pre-warmed liquid BHI and grown at 37 °C to an OD$_{578nm}$ 3.0 with 200 rpm shaking. Cultures were crosslinked with UV irradiation and immediately harvested. Cells were lysed and debris clarified by centrifugation. Supernatants were added to equilibrated M2 anti-FLAG resin (Sigma) and incubated at 4 °C with gentle rotation for 2 h. Resin-bound RNase III.RNA complexes were washed and underwent GST.TEV protease digestion at 18 °C with gentle rotation for 2 h. Eluates were collected by filtration through a Bio-spin chromatography column (Bio-Rad) and limited RNase digestion was performed using 0.15 U of RNace-IT (Agilent) to trim back overhang RNA. Eluates were added to equilibrated Ni-NTA resin and incubated at 4 °C with gentle rotation for at least 16 h. Resin-bound RNase III.RNA complexes were washed and 5′ RNA phosphate groups removed using 8 U of thermosensitive alkaline phosphatase (TSAP, Thermo). RNA was radiolabelled using a combination of 20 U T4 polynucleotide kinase (T4 PNK, NEB) and 30 μCi $^{32}$P-ATP. Linker aptamers were ligated consecutively on the 5′ (unique barcoded index) and 3′ ends of RNA using 40 U of T4 ssRNA ligase I (NEB). $^{32}$P-labelled RNase III.RNA complexes were eluted and resolved on a NuPAGE 4–12% gradient Bis-Tris PAGE gel (Invitrogen). Radiolabelled RNase III.RNA complexes were visualised by autoradiography and gel excised. Gel pieces were fragmented, recovered, and digested using 100 μg of Proteinase K at 55 °C for 2 h. RNA was phenol:chloroform extracted and cDNA libraries were constructed by performing reverse transcription and PCR amplification. Amplicons were separated on a 1.5% metaphor agarose gel and excised using the MiniElute gel extraction kit (Qiagen). Libraries were pooled, quantified on an Agilent Bioanalyzer, and sequenced using Illumina NextSeq500 platforms. Sequence data has been deposited at NCBI GEO under accession number GSE158830.

**Analysis of CLASH dataset.** Analysis of the CLASH data was similar to those described in Waters et al. (2017) and Tree et al. (2018)[13,64]. In brief, merging of pair-end reads was performed using 'bbmerge'[65]. Demultiplexing of samples was performed using 'pyCRAC'[66]. Quality and adapter trimming was performed using 'bbduk'[65]. De-duplication of PCR amplified reads was performed using 'pyCRAC'. Identification of the two RNAs that form the sequence hybrid representing the RNA–RNA interaction was performed using 'Hyb'[31]. The read counts corresponding to each RNA species from the hybrid sequences was identified using 'pyCRAC'. Statistical analysis of the significance of the RNA–RNA interactions was calculated using custom R scripts as described in Waters et al. (2017)[13]. The pipeline for CLASH data analysis, called Hyb-CRAC-R, has been implemented in Snakemake workflow management language and is available in the following GitHub repository (https://github.com/IgnatiusPang/Hyb-CRAC-R). Detailed protocols for the analysis of RNase III-CLASH data are presented in Supplementary Methods.

**Vancomycin spot dilution assay.** *S. aureus* JKD6009, JKD6008, and JKD6008 containing pSD1 constructs were used to inoculate 5 mL of liquid MH, supplemented with 10 μg/mL chloramphenicol where appropriate and grown 16 h at 37 °C with 200 rpm shaking. Cultures were aliquoted into a 96-well microtitre plate, serially diluted (up to $10^{-7}$) into liquid MH, and spotted onto solid MH plates supplemented with 100 ng/mL anhydrotetracycline (aTC), with and without 3 μg/mL vancomycin. Spot plates were air dried at room temperature and

incubated at 37 °C for 24 h. Plates were imaged on a Bio-Rad Chemi-doc using the default trans-white light setting.

**Construction of GFP translational fusions.** Target mRNAs *spoVG* and *mgrA* were amplified from JKD6008 gDNA (oligonucleotides listed in Supplementary Data 4) using Phusion Hot Start Flex DNA Polymerase and cloned into pCN33 by switching out the *gyrB* insert between the BglII and EcoRV sites. As the exception, the *isaA* mRNA was synthesised (Integrated DNA Technologies) to include the TSS, 5′UTR and exclude amino acids 2–29 as this incorporated the predicted signal peptide region. This was cloned into the BglII/EcoRV sites of pCN33. The P$_{tufA}$ promoter from JKD6008 was cloned at the PaeI/PstI sites of pICS3 to enable constitutive expression. RNAs encoding *sprX2*, *sprD*, *rsaA* and *vigR*$^{3′UTR}$ were cloned into the resulting pICS3::P$_{tufA}$ vector using the PstI/EcoRI sites (Supplementary Data 4). Electrocompetent *S. aureus* RN4220 was transformed with either the pCN33::P$_{tufA}$-mRNA::*gfp* or pICS3::P$_{tufA}$-sRNA constructs and stored at −80 °C as stationary phase cultures with 16% (v/v) glycerol. The transformed strains were then made electrocompetent and transformed a second time with the alternative vector, resulting in co-transformed cultures. These were then stored at −80 °C as stationary phase cultures with 16% (v/v) glycerol.

**Flow cytometry.** Co-transformed RN4220 GFP translation fusion constructs were streaked out onto solid BHI supplemented with 5 μg/mL erythromycin and 5 μg/mL chloramphenicol. RN4220 strain was chosen as it provided better fluorescence intensity within *S. aureus* clones. Individual colonies were used to inoculate 1 mL of 0.45 μm filtered liquid BHI and grown 16 h at 37 °C with 200 rpm shaking. Cultures were then diluted to an OD$_{578nm}$ 2.0 into 0.45 μm filtered PBS (pH 7.4) and at least 100,000 events were sampled on a LSRFortessa SORP (BD Biosciences) using a 530/30 nm bandpass filter. The PMT was adjusted so that background fluorescence was detectable and within range. The median fluorescence intensity (MFI) for the entire population for each test and control culture was determined using FlowJo software (version 8). The events were not gated and all events were recorded in the MFI. The MFI from *S. aureus* RN4220 (without GFP expressing plasmid) was used to determine background fluorescence and was subtracted from the MFI of test cultures. A two-sided Student's *t*-test assuming unequal variance was used to determine significance.

**Quantitative real-time PCR (qRT-PCR).** *S. aureus* JKD6008 and derivative cultures were diluted 1:100 in 10 mL fresh liquid BHI and grown at 37 °C with 200 rpm shaking to OD$_{578nm}$ 3.0. Cells were harvested by spinning at $3,800 \times g$ for 10 min at 4 °C. A total of 5 U of RNasin (Promega) and 10 U of RQ1 RNase-free DNase (Promega) was added and RNA purified using the GTC-phenol:chloroform extraction proceedure[60]. At least 1 μg of RNA was reverse-transcribed using SuperScript III (Thermo), according to the manufacturer's instructions. qRT-PCR was performed on a RotorGene Q (Eppendorf) using SensiFAST SYBR Hi-ROX mastermix (Bioline), according to the manufacturer's instructions. A total cDNA concentration of 50 ng in combination with 400 nM oligonucleotides were used per reaction (Supplementary Data 4). The Ct values per reaction were calculated using the RotorGene Q analysis software. Relative gene expression was determined using ΔΔCt abundance of the *gap* (SAA6008_RS08745, glyceraldehyde-3-phosphate dehydrogenase) transcript as a reference control[67].

**Growth curves.** JKD6008, JKD6009, and strain derivatives were plated onto solid MH supplemented with the appropriate antibiotics and incubated at 37 °C for 16 h. Single colonies were used to inoculate 5 mL of pre-warmed liquid MH and cultures incubated at 37 °C with 200 rpm shaking for a further 16 h. A volume of 300 μL of fresh liquid MH was added into each well of a sterile 100-well honey-comb microtiter Bioscreen plate (Thermo) in triplicates to give a starting OD$_{600nm}$ ~0.02. Vancomycin (2–3 μg/mL, where specified), teicoplanin (2 μg/mL), fosfomycin (2 μg/mL), ampicillin (5 μg/mL) and tigecycline (2 μg/mL) were added to assess sensitivity. Strains containing the pSD1 vector were also supplemented with 100 ng/mL aTC and 5 μg/mL chloramphenicol. The plate was incubated and analysed with a Bioscreen C spectrophotometer (Growth Curves USA) at 37 °C for 20 h with continuous low shaking, measuring OD$_{600nm}$ at 20 min intervals. Growth curves were obtained by plotting biological triplicates. The DMFit (DM: Dynamic Modelling, version 3.5) growth curve modelling software[68] was used to obtain values for the lag phase, growth rate (μ) and maximum OD.

**Northern blot.** Total RNA was purified using the GTC-phenol:chloroform extraction method as above. At least 3 μg of RNA was treated with a 5:1 ratio of glyoxal denaturation mixture for 1 h at 55 °C. Denatured RNA was resolved on a 1.5% BPTE-agarose gel containing SYBR Green (Thermo) and run for ~1 h at 100 V in 1× BPTE buffer. Intact rRNA was confirmed on a Bio-Rad Chemi-doc and washed consecutively in 200 mL of 75 mM NaOH, 200 mL of neutralising solution (1.5 M NaCl and 500 mM Tris-HCl, pH 7.5), and 200 mL of SSC buffer (3 M NaCl and 300 mM sodium citrate, pH 7.0) for 20 min each. RNA was capillary transferred onto a Hybond-N + nylon membrane (GE Healthcare) and UV-crosslinked in a Stratagene Auto-Crosslinker with 1200 mJ UV-C. The membrane was equilibrated in Ambion ULTRAhyb hybridisation buffer (Thermo) for 1 h at 42 °C and then incubated with 10 pmol of 20 μCi γ$^{32}$P-ATP-labelled

probe (oligonucleotide probes are detailed in Supplementary Data 4) for 16 h at 42 °C. Membranes were washed three times in 2× sodium chloride sodium phosphate EDTA (SSPE) buffer with the addition of 0.1% SDS for 15 min at 42 °C and imaged using a BAS-MP 2040 phosphorscreen on a FLA9500 Typhoon (GE Healthcare). For CRISPRi sRNA northern blots (Supplementary Fig. 3A) at least 2 µg of total purified RNA was resolved on an 8% polyacrylamide TBE-urea gel and transferred onto a nylon membrane for 16 h at 30 V in 0.5× TBE. The membrane was then crosslinked in a Stratagene Auto-Crosslinker with 1200 mJ UV-C and treated as identical to above.

**Transmission electron microscopy.** *S. aureus* JKD6009, JKD6008, and JKD6008 derivative constructs were streaked onto Columbia horse blood agar and grown at 37 °C for 16 h. Colonies were scraped from the agar and resuspended in 1 mL of PBS. Cultures were then centrifuged (50 × *g* for 5 min) and pellets resuspended in paraformaldehyde fixation solution and prepared for TEM as described previously by Howden et al.[69]. Cells were viewed and imaged on a FEI Tecnai G2 20 microscope at 15,000-22,000× magnification, with specific images taken at >100,000× magnification to focus on the cell wall. To determine cell wall thickness, 100 measurements of individual horizontally-planar cells were recorded using ImageJ and the mean and SEM reported. A two-sided Student's *t*-test assuming unequal variance was used to determine significance.

**RNA-seq and analysis of nucleotide data.** *S. aureus* JKD6008, JKD6008 ΔvigR, JKD6008 pSD1, and pSD1-*vigR*^3′UTR were grown in liquid MH with and without vancomycin (2 µg/mL) to an $OD_{600nm}$ 0.8. Strains containing the pSD1 vector were grown in the presence of 100 ng/mL of aTC and 5 µg/mL chloramphenicol. Cultures were harvested by centrifugation (3,900 × *g* for 10 min) and RNA purified using the GTC-phenol:chloroform protocol. This was repeated for three independently performed experiments, generating a total of 24 RNA samples. At least 3 µg of purified RNA underwent rRNA depletion, library preparation and then were sequenced on either an Illumina HiSeq150 platform generating 300-cycle paired-end reads (Novogene Inc, Hong Kong) or Illumina NextSeq500 generating 150-cycle single-end reads at the Ramaciotti Centre for Genomics (University of New South Wales, Sydney). The received demultiplexed sequence reads underwent quality control analysis and filtering using FastQC software. Poor quality nucleotides were trimmed using the trimmomatic software tool (v0.38) to reveal high-quality reads of at least 50 bp. Reads were aligned to the JKD6008 genome (Genbank NC_017341.1) using Bowtie2 (v2.3.4.2), allowing for 1 mismatch. The frequency of reads in each generated SAM file that mapped to the "gene" or "sRNA" feature of the JKD6008 genome GFF was calculated using HTseq-count[70]. The tabular datasets were used as input into DESeq2[71] to compare differential expression between biological replicate conditions. Any feature below a read count of 5 was filtered out using DESeq2 before differential expression analyses. The complete list of all differentially expressed genes and sRNAs are available at NCBI GEO under accession number GSE158830. Clusters of orthologous (COG) classes detailed for differentially expressed transcripts found in both *vigR*^Δ3′UTR and CRISPRi knockdown strains (adjusted *p* < 0.05) were determined using the protein function annotation by COG on WebMGA[72].

**RNA–RNA electrophoretic mobility shift assay.** Full-length or sub-fragments of *isaA* and *folD* mRNA, and *vigR* 3′UTR were in vitro transcribed (IVT) using 40 U of T7 RNA polymerase (Roche). IVT products were RQ1 DNase treated (Promega) for 15 min at 37 °C, phenol-chloroform extracted and ethanol precipitated, and then separated on a 6% polyacrylamide TBE-6M urea gel. Products were excised, crushed, and incubated in 500 µL RNA gel elution buffer (10 mM magnesium acetate, 0.5 M ammonium acetate, 1 mM EDTA) for 16 h at 4 °C. RNA was extracted from the eluate using phenol-chloroform extraction and ethanol precipitation. Approximately 50 pmol of RNA was dephosphorylated using Quick calf intestinal alkaline phosphatase (CIP, Thermo), then extracted using phenol-chloroform and ethanol precipitation. The 5′ ends of the RNA were radiolabelled with γ^32P-ATP using T4 polynucleotide kinase (NEB). Radiolabelled products were separated from free nucleotides using a MicroSpin G-50 column (Cytiva), and purified on a denaturing PAGE gel as above. To analyse *vigR* binding to *folD*, increasing excess amounts of *vigR* 3′UTR were annealed to 50 fmol of either radiolabelled full-length *folD* mRNA or a sub-fragment of *folD* in 1× duplex buffer (40 mM Tris-acetate, 0.5 mM magnesium acetate, 100 mM NaCl) in a 10 µL reaction. These were incubated at 95 °C for 5 min, then at 37 °C for 2 h. Samples were run on a 4% polyacrylamide 0.5X TBE gel containing 5% glycerol for ~4 h at a maximum of 16 V/cm or 1.33 mA/cm. Gels were then dried and visualised using a Fuji BAS-MP 2040 phosphorscreen and Typhoon FLA9500. To analyse *vigR* binding to *isaA*, increasing excess amounts of full-length *isaA* mRNA or sub-fragments of *isaA* were added to 25 fM of radiolabelled *vigR* 3′UTR in 1× duplex buffer. Where appropriate 1.25 µM of antisense competitor oligonucleotides (Supplementary Data 4) were added to compete away either *isaA* or radiolabeled *vigR* 3′UTR at a concentration excess of 500×. RNA was annealed, run, and visualised as above.

**Primer extension assays.** Approximately 100 µM of DNA oligonucleotide "vigR.3′ UTR.SHAPE" (Supplementary Data 4) was radiolabelled with γ^32P-ATP using T4 polynucleotide kinase (NEB) for 1 h at 37 °C. Radiolabelled products were separated from free nucleotides using a MicroSpin G-50 column (Cytiva), and purified on a 10% denaturing PAGE gel as above. Gel purified, in vitro transcribed *vigR* 3′ UTR and *isaA* Frag-C (5 pmol) were renatured by heating to 95 °C for 5 min and *vigR* 3′UTR was then incubated with (250× and 500× excess) and without *isaA* Frag-C in 1× duplex buffer as above, in the presence of 1 µg of yeast RNA (Invitrogen). Following incubation at 37 °C for 2 h, 0.2 U of RNase T1 (Invitrogen) and 0.4 U of ShortCut RNase III (NEB) were added and incubated at 25 °C for 10 min and 37 °C for 3 min, respectively. Reactions were stopped by placing on ice and performing phenol-chloroform extraction followed by ethanol precipitation. The pellet was hydrated in nuclease-free water and underwent reverse-transcription using SuperScript SSIV (Thermo), according to the manufacturer's instructions. Approximately 2.5 µM of the radiolabelled DNA oligonucleotide was used per reaction. Sequencing ddNTP ladders were obtained by reverse-transcription of 2 pmol of denatured *vigR* 3′UTR RNA with either 0.3 mM ddCTP, 0.9 mM ddTTP or 0.1 mM ddGTP termination mix and 2.5 µM of radiolabelled oligonucleotide. Following reverse-transcription, 1 µL of 4 M NaOH was added and incubated at 70 °C for 10 min to hydrolyse and degrade template RNA. Loading dye was added and samples denatured at 90 °C for 5 min and separated on a 4% polyacrylamide TBE-6M urea gel in 1× TBE buffer for ~2 h at a maximum of 40 W. Gels were dried and analysed using a Fuji BAS-MP 2040 phosphorscreen and Typhoon FLA9500.

**Statistics and reproducibility.** All Northern blots and EMSA analyses were performed a minimum of twice and the images presented here are representative of results from replicate experiments.

**Reporting summary.** Further information on research design is available in the Nature Research Reporting Summary linked to this article.

## Data availability

RNA-sequencing data generated in this study have been deposited as NCBI GEO under accession number GSE158830. Source data are provided with this paper.

## Code availability

Scripts used to analyse our Term-seq data are available at https://github.com/IgnatiusPang/TermPick. The snakemake pipline used to analyse our CRAC and CLASH datasets is available at https://github.com/IgnatiusPang/Hyb-CRAC-R.

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

## Acknowledgements

pICS3 and pCN33::P*tufA*-*gyrB*::*gfp* vectors were a kind gift from Brice Felden (Université de Rennes). The CRISPRi knockdown pSD1 vector was a kind gift from Baolin Sun (University of Science and Technology of China). S.G. was supported by Medical Research Council Non-Clinical Senior Research Fellowship (MR/R008205/1). S.W.M. was supported by the Wellcome Trust (109093/Z/15/A). D.G.M., B.M.S., and J.J.T. are supported by grants from the National Health and Medical Research Council, Australia (GNT1139313 and GNT1161161). S.W. and W.W. are supported by Research Training Programme Scholarships from the Australian Government.

## Author contributions

J.J.T. and T.P.S. initiated the project and D.G.M., J.L.W., M.R.W., B.P.H., T.P.S., S.G., and J.J.T. designed the experiments. D.G.M., J.L.W., W.G., S.M., C.N.I.P., S.W., W.W., B.S., I.R.M., and J.M.B. performed the experiments. D.G.M., J.L.W., C.N.I.P., W.W., and J.J.T. analysed the data. C.N.I.P. developed the pipeline and provided statistical analysis of CLASH data. S.G. developed the analysis pipeline for paired end CRAC data. D.G.M. and J.J.T. drafted the manuscript and all authors reviewed the manuscript and approved the final version.

## Competing interests

The authors declare no competing interests.
