## [Peer Review File · Nature Communications]

Reviewer comments, first round of review

Reviewer #1 (Remarks to the Author):

In their manuscript "RNase III-CLASH of multi-drug resistant *Staphylococcus aureus* reveals a regulatory mRNA 3'UTR required for intermediate vancomycin resistance", Mediati et al. apply the CLASH method, previously established in *E. coli* by the same lab, to methicillin-resistant *Staphylococcus aureus* (MRSA) to characterize the network of RNA-RNA interactions mediated by RNase III. They aim to identify post-transcriptional regulators that contribute, at least in part, to the appearance of vancomycin-intermediate tolerance in clinical strains. They identify a regulatory 3'UTR, named *vigR*-3'UTR, and suggest this region to be a hub for regulation of genes involved in vancomycin-intermediate tolerance. They can show that deletion of this 3'UTR causes higher sensitivity to vancomycin. The authors carry on with a gene expression analysis to identify genes de-regulated in a *vigR* 3'UTR knock out or knock down. They focus on *fold* and *isaA*, the latter of which has been known to be involved in cell wall architecture. By showing the influence of *isaA* in cell wall thickness, the authors propose that, at least in part, the 3'UTR of *vigR* promotes resistance to vancomycin by upregulation of the *isaA* gene product and therefore by increasing cell wall thickness.

With respect to the CLASH dataset, I have very little concerns, apart from a few more explanations that would better guide the readers through the analysis and the results (see comments below). The biological part and characterization of *vigR*-3'UTR will need more experimental verification, mostly including more controls and providing supporting evidence on the connection between the *vigR* phenotype and the *isaA* regulation (see detailed comments below).

Major comments:

- L195-197: This information cannot be easily extracted from the table because it lacks any gene names. I think highlighting those genes/hybrids in the table that are specifically mentioned in the main text would make the navigation easier for the reader
- Figure 2B: the graph suggests that the vast majority of interactions was detected in only one experiment/replicate? Please, comment on that.
- L210-211: The authors collated their hybrid reads with additional RNase III-CLASH data generated in a study submitted in parallel (McKellar et al.). Why was this done, and why was it done only for the RPMI medium condition? Please, explain briefly in the text. Further, does Suppl. Table 3 contain the collated data (please, also mention in the text)? Was the *vigR*-3'UTR-*isaA* interaction found in both CLASH conditions?
- I find it confusing that the authors refer to the *vigR* 3'UTR as an sRNA. I would prefer if they avoid the term as it is misleading. I strongly prefer their choice in the title where they refer to it as a regulatory 3'UTR
- Given the large transcript size of sRNA275, its location in the 3UTR of E0E12_RS09390 mRNA and the lack of a 5' processing site that would release the sRNA from the parental mRNA, it is unclear to me why it was originally classified as an sRNA. Please, explain briefly.
- Please spend a few more lines and maybe some schematic representation to explain the *vigR* mutants. What is a *vigR* 3UTR repaired? Along this line and referring to Fig 4C, there is no band corresponding to the *vigR* CDS in the blot. Additionally, I would have expected a shorter band for the *vigR* 3UTR deletion. Is that because the probe was designed to hybridize in the 3UTR? In that case, it'd be good to have a probe binding in the CDS region to show the stability of the truncated isoform is not affected by this mutation.
- Figure 4B and Suppl. Figure 3B: the *vigR* knockdown seems to have a growth defect (or at least lower max. OD600) even without vancomycin. How does that influence MIC determination?
- Please label the figures according to the names you use in the main text: e.g., pSD1-*vigR*3'UTR in Fig. 4B corresponds to sRNA275 knock down in the main text; or VSSA strain in Fig. 4C corresponds to JKD6009 in the text. It is really hard to keep up with it.
- How many of the differentially regulated genes from the RNA-seq of *vigR* 3UTR mutant/CRISPR knockdown were found in the CLASH? Is the deletion/CRISPR knockdown also affecting *vigR* CDS? The authors should comment on this.
- The EMSAs in Fig. 6F show Kd values in the micromolar range. Is this physiologically relevant?

From the very long predicted duplex I would have expected a higher affinity.

- Additionally, Fig. 6C to 6H could be reduced to one single panel (or be nicely complemented) by performing structure probing. This would show specificity of the binding and would also show the exact interaction site between the two RNAs in the context of the full-length transcript.
- What is the role of RNase III in the identified interactions? It seems like the binding of vigR-3'UTR to isaA stabilizes the target mRNA, but wouldn't it be expected to be the opposite as RNase III would cut upon interaction? Please, briefly respond to this in the main text.

Minor comments:

- Supplementary table 3: the table descriptions are a bit confusing. What exactly is the difference between the first and second table?
- Fig. 2H: the murQ/RNAIII predicted interaction seems to be too short for being RNase III bound, but still it was detected in the dataset. Is there a possibility the CLASH could detect protein-unbound duplexes as well?
- Figure 3: This figure could be improved to make it more informative, e.g., one could highlight (label) some of the interactions that are specifically mentioned in the main text. Are the sRNA-RNA interactions identified clustering in a particular way (e.g., similar to what was done for the RNA-seq data in Fig. 5B)?
- L216: "...media conditions ..." means BHI and RPMI?
- L219: A reference for SprD function is missing
- Figure 2G: Please, fully describe the value plotted on the y-axis. Is this a ΔCt or a $\Delta\Delta Ct$ value?
- L237: The authors mention "... our RNase III-CLASH network..." Do they refer to their own data or is this the collated dataset?
- L258: I think the conclusion that sRNA275 expression is increased in JKD6008 is not warranted because the ribosomal controls also show stronger signals. Please, comment.
- L276 and Figure 4E: The authors state that the vigRCDS deletion has a slight growth defect in MH medium but the effect seems extremely subtle. Please, support this statement by quantifying maximum OD600 values and/or growth rates.
- L313: Please state the fold change of the fold and isaA in the vigR backgrounds, it is not easy to extrapolate from the volcano plots in Fig. 5A and 5C. Please, also mention the (predicted) function of fold. Does it also have a role in cell wall metabolism?
- L372-376 (Suppl. Figure 4D and E): Why would you conclude that vigR influences cell wall architecture when stating that the results from Suppl. Figure 4D and 4E were not significant? Please, consider qualifying your statement
- Figure 6J is missing the y-axis label

Reviewer #2 (Remarks to the Author):

Mediati and collaborators studied the sRNA regulatory networks using CLASH in the context of antibiotic resistance and especially last line antibiotics such as vancomycin. Because Hfq is dispensable for sRNA-mRNA interactions in this bacterium, the authors adapted CLASH using RNase III as a bait, based on the assumption that this double-strand endoribonuclease is involved and plays a key role in sRNA-mediated regulation. After an overall description of the CLASH data obtained in JKD6009 (a VSSA strain) and validation of CLASH, the authors pointed that significant amount of RNase III interactions occurred between RNAs and especially UTRs. Based on previous publications that identified sRNAs potentially involved in vancomycin tolerance and their CLASH analysis, the authors knockdown the expression of six putative sRNAs in JKD6008 (a VISA strain) and tested the tolerance to vancomycin. Among them, a VSSA phenotype was recovered from the knockdown of sRNA275, a mRNA that contain a long 3'UTR named vigR. The authors showed that the 3'UTR of the mRNA is responsible for glycopeptide tolerance and that the RNA stabilizes fold and isaA RNAs. Finally, they show that regulation of isaA has an impact on cell wall thickness which could be responsible for vancomycin-intermediate tolerance. Although the differences are sometimes modest in their subsequent experiments the manuscript describes a novel role of 3'UTR is post-transcriptional regulation and therefore bring an additional layer in the coordination of gene expression in *S. aureus*. However, some points need to be addressed to convince a large audience.

General comment:

Supplementary figure 3b. What about the growth of the two strains in the absence of vancomycin? This would help to appreciate the actual fitness of each strain without stress.

Figure 4c: The data presented are not convincing. There is no normalization and quantification on the northern blot. qPCR may be performed to get more precise results. On figure 4c, I cannot find the vigRdCDS mutant while it appears in the text.

Supplementary figure 3c. Quantification is lacking although it is more convincing overnight.

Figure 6A: How many replicates were done? As transcript variations are modest, statistical analyses must be performed.

Line 224: 'repression could be partially restored' rather than 'repression could be restored'.

Reviewer #3 (Remarks to the Author):

Review for "RNase III-CLASH of multi-drug resistant *Staphylococcus aureus* reveals a regulatory mRNA 3'UTR required for intermediate vancomycin resistance", Mediati et al.

Mediati et al. report the identification of a novel regulatory hub in *S. aureus*, VigR, which is involved in the resistance to vancomycin, a last resort antibiotic to treat MRSA. Several interesting new findings are being reported, especially around the unexpected prevalence of mRNA-mRNA interactions detected by RNase III CLASH. Previous research on gram-negative bacteria by the same group used RNase E CLASH to profile sRNA-mRNA interactions associated with endoribonuclease RNase E in pathogenic *E. coli*. Here, the CLASH protocol was adapted (RNase III CLASH) to gram-positive bacteria. Focusing on the clinically important antibiotic resistance of MRSA, the authors identified novel targets of VigR, namely *foiD* and *isaA*. The regulatory loop and the functional implications of these RNA interactions have been thoroughly validated using an impressive variety of different techniques and approaches. The finding that an mRNA rather than a protein is crucially involved in regulation of vancomycin resistance is of general interest. The application of dRNA-seq and Term-seq to complement CLASH allows the authors to identify RNA elements within RNA transcript that modulate the expression of CDS as well as novel regulatory sRNAs. Overall, a carefully conducted study with thoroughly validated data revealing interesting new aspects of RNA biology in bacteria.

Points to address:

- The study is a very interesting read but to broaden the readership beyond the bacterial community, a more general introduction into prokaryotic sRNA gene regulation pathways might be useful, also to be able to better understand the role of RNaseIII in the pathway.
- CLASH revealed 133 sRNA-mRNA interactions and 543 statistically significant mRNA-mRNA interactions suggesting that mRNAs may exert regulatory functions in trans. Can the authors speculate why in the given experimental setup coding mRNA-mRNA interactions seem to be more prevalent than sRNA-mRNA interactions?
- It is unclear why replicates 1-2 and 3,4,5,6 underwent different protocols: why 2 samples with one protocol and 4 samples with another and why using different protocols?
- In mammalian cells, qPCR is generally performed using 3 reference genes. Here, only 1 was used (*gap*). Is this a particularly stable gene under the conditions applied here or might 3 different reference genes be better?
- In the discussion, the authors could elaborate a bit more on the clinical implications their findings might have, i.e. targeting of RNA interactions as a therapeutic approach. Could antisense oligonucleotides be envisaged in the future to target such interactions?
- VigR 3'UTR is neither independently transcribed nor processed from the *vigR* mRNA. Hence, the authors conclude that *vigR* is a regulatory mRNA and is so far only the 3rd example of bacterial mRNA with trans-regulatory function. Is it possible that VigR is not transcribed under the chosen experimental conditions (medium, growth conditions)? It might be interesting to check for VigR transcription or processing under different conditions or in other bacterial strains.

REVIEWER COMMENTS

Reviewer #1 (Remarks to the Author):

In their manuscript “RNase III-CLASH of multi-drug resistant *Staphylococcus aureus* reveals a regulatory mRNA 3’UTR required for intermediate vancomycin resistance”, Mediati et al. apply the CLASH method, previously established in *E. coli* by the same lab, to methicillin-resistant *Staphylococcus aureus* (MRSA) to characterize the network of RNA-RNA interactions mediated by RNase III. They aim to identify post-transcriptional regulators that contribute, at least in part, to the appearance of vancomycin-intermediate tolerance in clinical strains. They identify a regulatory 3’UTR, named *vigR*-3’UTR, and suggest this region to be a hub for regulation of genes involved in vancomycin-intermediate tolerance. They can show that deletion of this 3’UTR causes higher sensitivity to vancomycin. The authors carry on with a gene expression analysis to identify genes down-regulated in a *vigR* 3’UTR knock out or knock down. They focus on *fold* and *isaA*, the latter of which has been known to be involved in cell wall architecture. By showing the influence of *isaA* in cell wall thickness, the authors propose that, at least in part, the 3’UTR of *vigR* promotes resistance to vancomycin by upregulation of the *isaA* gene product and therefore by increasing cell wall thickness.

With respect to the CLASH dataset, I have very little concerns, apart from a few more explanations that would better guide the readers through the analysis and the results (see comments below). The biological part and characterization of *vigR*-3’UTR will need more experimental verification, mostly including more controls and providing supporting evidence on the connection between the *vigR* phenotype and the *isaA* regulation (see detailed comments below).

Major comments:

- L195-197: This information cannot be easily extracted from the table because it lacks any gene names. I think highlighting those genes/hybrids in the table that are specifically mentioned in the main text would make the navigation easier for the reader

Thank you for this suggestion. We have now added gene names and sRNA names to Supplementary Table 3 for the hybrid interactions that are described in the text. We have also added common names and locus tags for all features where this information is available.

- Figure 2B: the graph suggests that the vast majority of interactions was detected in only one experiment/replicate? Please, comment on that.

The reviewer is correct, the majority of RNA-RNA interactions are only recovered in one experiment or replicate. This is consistent with our previous RNase E-CLASH dataset. We speculate that this represents shallow sampling of a large pool of RNA-RNA interactions that is dominated by a few highly abundant interactions.

We have updated the text at lines 205-209 to add a comment on these results:

“We recovered 13,530 unique hybrid reads (21,680 in the collated dataset), representing 822 statistically significant unique RNA-RNA interactions (1,420 in the collated datasets), including 133 sRNA-mRNA interactions (Supplementary Table 3). Consistent with our earlier dataset¹ many interactions are recovered in a single experiment with 117 interactions recovered in multiple independent CLASH experiments.”

- L210-211: The authors collated their hybrid reads with additional RNase III-CLASH data generated in a study submitted in parallel (McKellar et al.). Why was this done, and why was it done only for the RPMI medium condition? Please, explain briefly in the text. Further, does Suppl. Table 3 contain

the collated data (please, also mention in the text)? Was the *vigR*-3'UTR-*isaA* interaction found in both CLASH conditions?

Our aim for this study was to identify RNA interactions that are important for vancomycin tolerance in VISA. From this perspective, it did not make sense to ignore our collaborators parallel data that was generated using the same strains and protocol. We surveyed all available interactions and tested RNA-RNA interactions from this collated dataset to identify RNAs required for the vancomycin tolerance phenotype. We used both RPMI and TSB data in the collated dataset.

We have updated the text to clarify the datasets used for the collated dataset:

“We collated our hybrid reads with additional RNase III-CLASH data generated in a parallel study by MacKellar *et al* (submitted with this manuscript) utilising TSB and RPMI-1640 media (Supplementary Table 3).”

Supplementary Table 3 contains the collated RNase III-CLASH dataset and this table can be sorted by the reader based on the specific CLASH experiment. This has also been made clearer in the text by specifying Supplementary Table 3 in parentheses when introducing the collated CLASH data: “the cumulative *S. aureus* sRNA interactome contains 287 nodes and 256 sRNA interactions (Figure 3 and Supplementary Table 3)”.

- I find it confusing that the authors refer to the *vigR* 3'UTR as an sRNA. I would prefer if they avoid the term as it is misleading. I strongly prefer their choice in the title where they refer to it as a regulatory 3'UTR

Thank you for this comment. We have updated the text to remove references to *vigR* 3'UTR as a sRNA – we now use the term regulatory RNA. We also propose the new term “*vigR* 3'UTR” for sRNA275 earlier in the manuscript than previously.

- Given the large transcript size of sRNA275, its location in the 3UTR of E0E12_RS09390 mRNA and the lack of a 5' processing site that would release the sRNA from the parental mRNA, it is unclear to me why it was originally classified as an sRNA. Please, explain briefly.

The sRNA275 transcript was identified by manual curation of RNA-seq data in Howden *et al* AAC 2013. The RNA-seq data is relatively low coverage (between 71,000 and 690,000 reads per sample) and it's likely that the combination of permissive criteria (the authors identified 357 sRNAs) and poor coverage led to the incorrect annotation of the *vigR* mRNA 3'UTR.

- Please spend a few more lines and maybe some schematic representation to explain the *vigR* mutants. What is a *vigR* 3UTR repaired? Along this line and referring to Fig 4C, there is no band corresponding to the *vigR* CDS in the blot. Additionally, I would have expected a shorter band for the *vigR* 3UTR deletion. Is that because the probe was designed to hybridize in the 3UTR? In that case, it'd be good to have a probe binding in the CDS region to show the stability of the truncated isoform is not affected by this mutation.

The Northern probe is designed against sRNA275 (*vigR* 3'UTR). We have now added a schematic representation of the *S. aureus* *vigR* constructs to Supplementary Figure 3F and indicate the position of the Northern blot probe used to characterise *vigR*.

The text has been updated to describe the mutants and reference Supp Fig 3F (lines 291-295):

“To determine the relative contribution of each region to intermediate-vancomycin tolerance, clean deletions of both the 3'UTR (*vigR*^{Δ3'UTR}) and CDS (*vigR*^{ΔCDS}), and a chromosomally repaired *vigR*^{Δ3'UTR} (*vigR*^{Δ3'UTR}-repair, restoring the wild type genotype) were constructed in JKD6008 (schematic representation of constructs in Supplementary Figure 3F).”

We have also performed qRT-PCR to assess the stability of both the *vigR* 3'UTR and CDS in the deletion constructs and repair construct. We show that stability of the CDS is moderately affected by the 3'UTR deletion (CDS transcript levels 38% in the *vigR*^{Δ3'UTR} and restored to 81% in the *vigR*^{Δ3'UTR}-repair relative to WT, $p=0.012$) (Supplementary Figure 3Gi). We also verify that in the *vigR*^{ΔCDS} strain, the abundance of the 3'UTR is moderately stable ($69\% \pm 6.8$; Supplementary Figure 3Gii). This is likely contributing to the slight growth attenuation of the *vigR*^{ΔCDS} strain (related to a previous comment below).

The following text has been added:

“These strains were confirmed using Northern blot analysis (Figure 4D), qRT-PCR (Supplementary Figure 3G) and whole genome sequencing. We find that the *vigR* 3'UTR is required for *vigR* CDS stability (CDS transcript levels are 37.5% c.f. WT, Supplementary Figure 3Gi). The 3'UTR of *vigR* is more stable in the absence of the CDS (68.2% c.f. WT; Supplementary Figure 3Gii).

- Figure 4B and Suppl. Figure 3B: the *vigR* knockdown seems to have a growth defect (or at least lower max. OD600) even without vancomycin. How does that influence MIC determination?

We have now performed statistical analysis on the growth curves for the *vigR* knockdown (pSD1-*vigR*) in the presence and absence of vancomycin and have compared this to the parent construct (pSD1) (Supplementary Figure 3C).

We confirm that the pSD1-*vigR* construct has an 18.4% lower max OD than the pSD1 parent strain in the absence of vancomycin. However, in the presence of vancomycin, the pSD1-*vigR* construct has a 31.6% lower max OD than pSD1. We confirm the pSD1-*vigR* is sensitive to vancomycin treatment when compared to growth in the absence of vancomycin (~1.25-fold decrease in max OD, Pvalue=0.00054). In comparison, the max OD of the pSD1 parent construct +/- vancomycin is not statistically different, confirming vancomycin has very little effect on the max OD of the pSD1 parent strain.

In the presence of vancomycin, the pSD1-*vigR* construct has a ~1.53-fold increased lag phase than pSD1 (Pvalue=0.0075). We confirm the pSD1-*vigR* construct has a ~1.91-fold increase in the lag phase (Pvalue=0.0012) in the presence of vancomycin when compared to growth in the absence of vancomycin. In comparison, the lag phase of the pSD1 parent construct +/- vancomycin is not statistically different (Pvalue=0.15), confirming vancomycin has very little effect on the lag phase of the pSD1 parent strain.

We have added these analyses into the text and Supplementary Figure 3C. Although there is a slight growth defect of the pSD1-*vigR* knockdown in the absence of vancomycin, in the presence of vancomycin and when compared to the pSD1 parent construct the lag phase is significantly increased and the max OD is significantly decreased, indicating minimal influence on MIC measurements.

- Please label the figures according to the names you use in the main text: e.g., pSD1-*vigR*3'UTR in Fig. 4B corresponds to sRNA275 knock down in the main text; or VSSA strain in Fig. 4C corresponds to JKD6009 in the text. It is really hard to keep up with it.

Thank you for highlighting this. We have added the new name to the first mention of sRNA275 for clarity and amended Figure 4B and 4D to include both terms. Our new name is introduced in the text with Figure 4 and we use *vigR* from this point forward.

Lines 259-261:

“However in the presence of a sub-inhibitory concentration of vancomycin (3 mg/mL), growth of the

regulatory RNA knockdown annotated as sRNA275 (here termed *vigR* 3'UTR) was reduced 1000-fold (Figure 4Aii)."

- How many of the differentially regulated genes from the RNA-seq of *vigR* 3UTR mutant/CRISPR knockdown were found in the CLASH? Is the deletion/CRISPR knockdown also affecting *vigR* CDS? The authors should comment on this.

The *vigR* CLASH targets *fold* and *isaA* were the only targets differentially expressed in our RNA-seq analysis (highlighted in Figure 5A).

The CRISPRi knockdown targeting the *vigR* 3'UTR is expected to repress the *vigR* CDS which is reduced in the RNA-seq data ($\log_2FC \sim -1.6$, FDR=0.00041). We have now performed qRT-PCR to assess the stability of the *vigR* CDS in the 3'UTR deletion and repair construct. We show that stability of the CDS is moderately affected by the 3'UTR deletion (CDS transcript levels 38% in the *vigR* ^{Δ 3'UTR} and restored to 81% in the *vigR* ^{Δ 3'UTR}-repair relative to WT, $p=0.012$) (Supplementary Figure 3Gi).

This has now been commented on in the text (Lines 295-298) and discussed above in regards to a previous comment.

- The EMSAs in Fig. 6F show Kd values in the micromolar range. Is this physiologically relevant? From the very long predicted duplex I would have expected a higher affinity.

We have used relatively long RNAs for these EMSAs (*vigR* 3'UTR=650 nts, *isaA*=800 nts) and the relatively high Kd may reflect structure within both RNAs. We do not see a shift at similar concentrations of *isaA* mRNA fragment B (Supp Figure 6B) indicating that the interaction is specific. We are also able to compete the interaction away using a 20-mer indicating that the interaction requires nucleotides from +1069-1038 of *vigR* and complementary nucleotides at +645-656 of *isaA* fragment C (Figure 6H).

Kd values in the micromolar range have been previously reported with long structured 3'UTRs in *S. aureus* - Ruiz de los Mozos *et al.* showed that the *icaR* 5' and 3'UTRs directly interact in the 1.4 μ M range (Ruiz de los Mozos *et al.* 2013 PLoS Genet). Additionally Gerovac *et al.* showed that the FopA and Inc RNA species have an affinity constant of $\sim 1 \mu$ M in *Salmonella* (Gerovac *et al.* 2020 RNA). Micromolar concentrations were also required to demonstrate an interaction between the *Salmonella* sRNA PinT and the mRNA *steC* (Santos *et al.* 2021 Cell Reports) indicating that *in vitro* interactions in the micromolar range can be physiologically relevant.

Importantly, these *in vitro* assays are performed in the absence of any protein chaperones and the RNAs may anneal at lower concentrations in the cell with the appropriate chaperones.

- Additionally, Fig. 6C to 6H could be reduced to one single panel (or be nicely complemented) by performing structure probing. This would show specificity of the binding and would also show the exact interaction site between the two RNAs in the context of the full-length transcript.

We have now performed structure probing on the *vigR-isaA* fragment C interaction and probed the interaction site using ShortCut RNase III that cleaves dsRNA (new Supplementary Figure 6C and 6D). Consistent with our results obtained with antisense competitors, *vigR* is cleaved by ShortCut RNase III at position C+1063 (at antisense oligo C1) in the presence of *isaA* frag C. This result supports duplex formation between *vigR* and *isaA* fragment C.

- What is the role of RNase III in the identified interactions? It seems like the binding of *vigR*-3'UTR to *isaA* stabilizes the target mRNA, but wouldn't it be expected to be the opposite as RNase III would cut upon interaction? Please, briefly respond to this in the main text.

We have now constructed an *rnc* (RNase III) deletion in JKD6009 (VSSA). Attempts to construct the *rnc* deletion in the JKD6008 (VISA) background were unsuccessful. We find that deletion of *rnc* increases the abundance of *isaA* transcript, suggesting that *vigR* 3'UTR protects *isaA* by blocking RNase III processing of an *isaA* stem-loop or dsRNA secondary structure.

We have not resolved the mechanism of activation as yet but (as presented in the discussion) we note that both the *hly-prsA* and *irvA-gbpC* mRNA-mRNA interactions protect the target mRNA from RNase attack (refs 43 and 44). We suggest that *vigR* may also protect *isaA* from ribonuclease attack.

In our previous study using RNase E to capture RNA-RNA interactions we also identified activating sRNA-mRNA interactions. Specifically, our recent follow-up paper focussed on the activating interaction between StxS sRNA and *rpoS* mRNA that was abundantly crosslinked to RNase E (Sy *et al* PNAS 2020). We propose two plausible explanations, 1) even activating regulatory RNA-mRNA interactions are ultimately turned over in the cell and associate with the RNA degradation machinery where they can be captured by RNase-CLASH, or 2) that the RNA-binding domains of the RNA degradosome may also play roles in facilitating RNA-RNA interactions that are not always transferred to the catalytic site. We speculate that the flexible RBD domains of RNase III may facilitate RNA-RNA interactions that are not cleaved at the catalytic site.

Minor comments:

- Supplementary table 3: the table descriptions are a bit confusing. What exactly is the difference between the first and second table?

Supplementary Table 3 (Tab 1, all hybrids) includes interactions that were merged together across multiple experiments (where column 1 indicates “Merged”). Information regarding the number of hybrid reads from each experiment, which experiments had the hybrid reads, and the FDR of interactions from individual experiments, are lost in these merged clusters.

To ensure that this information is available, we have included a separate table that provides all of the additional details from the “merged” interactions. This is provided in Tab 2 “hybrids before merging”.

- Fig. 2H: the *murQ*/RNAIII predicted interaction seems to be too short for being RNase III bound, but still it was detected in the dataset. Is there a possibility the CLASH could detect protein-unbound duplexes as well?

Thank you for this question – it touches on an important point regarding RNase III substrates. The Romby lab have demonstrated that RNase III in *Staphylococcus aureus* processes co-axially stacked helices that interact through kissing loop interactions for the sRNA-mRNA pairs *rot*-RNAIII (7nt loop-loop interaction) and *coa*-RNAIII (6nt loop-loop interaction) (Romilly *et al* RNA Biology 2012; Biosset *et al* Genes and Dev, 2006).

We speculate that short interactions (like *murQ*-RNAIII) may be adopting similar co-axially stacked RNAs stems loops that interact through short loop-loop interactions to create co-axially stacked stems that form RNase III binding substrates. This highlights the structural flexibility of RNase III substrate recognition.

- Figure 3: This figure could be improved to make it more informative, e.g., one could highlight (label) some of the interactions that are specifically mentioned in the main text. Are the sRNA-RNA interactions identified clustering in a particular way (e.g., similar to what was done for the RNA-seq data in Fig. 5B)?

Thank you for this suggestion. We have performed GO and COG enrichment analysis on the mRNA targets for each sRNA that was identified in RNase III-CLASH, analogous to previous analysis that was performed for RNase E-CLASH in *E. coli* (Waters *et al.* 2017 *EMBO J*). Unfortunately, we did

not find enrichment of ontological classes of the sRNA targets – potentially reflecting the limited detail in the annotation of our clinical VSSA isolate.

We have updated Figure 3 to include key RNA-RNA interactions.

- L216: “...media conditions ...” means BHI and RPMI?

“Media conditions” has been changed to “BHI or RPMI-1640 media conditions” to be more specific.

- L219: A reference for SprD function is missing

The Chabelskaya *et al.* 2010 *PLoS Pathog* reference has been added.

- Figure 2G: Please, fully describe the value plotted on the y-axis. Is this a ΔCt or a $\Delta\Delta\text{Ct}$ value?

This is relative $\Delta\Delta\text{Ct}$ abundance. The following has been included in the Methods Section “Relative gene expression was determined using $\Delta\Delta\text{Ct}$ abundance of the *gap* transcript as a reference control.” The y-axis of Figure 2G and 2H has also been changed to include “ $\Delta\Delta\text{Ct}$ abundance”.

- L237: The authors mention “... our RNase III-CLASH network...” Do they refer to their own data or is this the collated dataset?

This sentence refers to the collated network and has been changed to “our collated RNase III-CLASH network”.

- L258: I think the conclusion that sRNA275 expression is increased in JKD6008 is not warranted because the ribosomal controls also show stronger signals. Please, comment.

We agree that the ribosomal controls also show stronger signals in JKD6008 and in line with Reviewer 3’s comment we have quantified the *vigR* transcript expression and have normalised to the ribosomal RNA controls. This quantification is reported within the figure and supports a ~20% increase in expression of *vigR* in JKD6008 when compared to JKD6009. This is also consistent with Northern blot quantification analysis in Supplementary Figure 3D for BHI media – in line with Reviewer 3.

- L276 and Figure 4E: The authors state that the *vigR*CDS deletion has a slight growth defect in MH medium but the effect seems extremely subtle. Please, support this statement by quantifying maximum OD600 values and/or growth rates.

The log (growth rate) phase and maximum OD600 has been quantified using the DMFit software for all growth curves in Figure 4 and results are presented in Supplementary Figure 4A. In MH media (no antibiotics), the quantified results show:

1. Slight attenuation in the log rate of the *vigR* CDS deletion when compared to WT (A mean decrease of 19% relative to WT, $p=0.002$).
2. Slight attenuation in the max OD of the *vigR* CDS deletion when compared to WT (A mean decrease of 7% in the max OD compared to WT, $p=0.01$).

There is little change when comparing the corresponding growth phases in the presence of antibiotics. The following has been added to the Methods: “The DMFit (DM:dynamic Modelling, version 3.5) growth curve modelling software was used to obtain values for the lag phase, growth rate and maximum OD.

- L313: Please state the fold change of the *fold* and *isaA* in the *vigR* backgrounds, it is not easy to extrapolate from the volcano plots in Fig. 5A and 5C. Please, also mention the (predicted) function of *fold*. Does it also have a role in cell wall metabolism?

We have included into the text “The *folD* mRNA (reduced in the knockdown strain, FDR=0.0047, log₂FC=-1.21) involved in folate metabolism, produces tetrahydrofolate which subsequently forms as a key metabolite for amino acid (histidine) and nucleotide (purine) biosynthesis.”

- L372-376 (Suppl. Figure 4D and E): Why would you conclude that *vigR* influences cell wall architecture when stating that the results from Suppl. Figure 4D and 4E were not significant? Please, consider qualifying your statement

We have removed *vigR* 3'UTR from our concluding statement in this section and updated the language used when describing the TEM results for the *vigR* 3'UTR deletion strain.

Lines 412-415: “Interestingly cell wall thickness measurements of the *vigR*^{Δ3'UTR} strain revealed a decrease in cell wall thickness to 24.13 nm ($p=0.058$) when compared to the isogenic VISA parent strain, suggesting that *vigR* 3'UTR influences cell wall architecture in *S. aureus* (Supplementary Figure 7C and 7D).”

- Figure 6J is missing the y-axis label

The following has been added to the y-axis “Cell wall length (nm)”

Reviewer #2 (Remarks to the Author):

Mediati and collaborators studied the sRNA regulatory networks using CLASH in the context of antibiotic resistance and especially last line antibiotics such as vancomycin. Because Hfq is dispensable for sRNA-mRNA interactions in this bacterium, the authors adapted CLASH using RNase III as a bait, based on the assumption that this double-strand endoribonuclease is involved and plays a key role in sRNA-mediated regulation. After an overall description of the CLASH data obtained in JDK6009 (a VSSA strain) and validation of CLASH, the authors pointed that significant amount of RNase III interactions occurred between RNAs and especially UTRs. Based on previous publications that identified sRNAs potentially involved in vancomycin tolerance and their CLASH analysis, the authors knockdown the expression of six putative sRNAs in JKD6008 (a VISA strain) and tested the tolerance to vancomycin. Among them, a VSSA phenotype was recovered from the knockdown of sRNA275, a mRNA that contain a long 3'UTR named *vigR*. The authors showed that the 3'UTR of the mRNA is responsible for glycopeptide tolerance and that the RNA stabilizes fold and *isaA* RNAs. Finally, they show that regulation of *isaA* has an impact on cell wall thickness which could be responsible for vancomycin-intermediate tolerance. Although the differences are sometimes modest in their subsequent experiments the manuscript describes a novel role of 3'UTR is post-transcriptional regulation and therefore bring an additional layer in the coordination of gene expression in *S. aureus*. However, some points need to be addressed to convince a large audience.

General comment:

Supplementary figure 3b. What about the growth of the two strains in the absence of vancomycin? This would help to appreciate the actual fitness of each strain without stress.

Thank you for the comment. We have now quantified and performed statistical analysis on growth phases for the *vigR* knockdown (pSD1-*vigR*) and parent construct (pSD1) in the absence of vancomycin (see previous comment from reviewer #1). These analyses are presented in Supplementary Figure 3C.

The *vigR* CRISPRi knockdown strain does have a reduced log (μ) growth rate (31.4% decrease, $p=0.0036$) and this is not changed in the presence of sub-inhibitory vancomycin. The lag phase is similar in the control and *vigR* knockdown without vancomycin. The maximal OD in the *vigR* knockdown is reduced 18.3% ($p=0.0047$) relative to the pSD1 without vancomycin. However, in the presence of vancomycin the lag phase becomes more pronounced in the *vigR* knockdown (1.53-fold increase c.f. vector control, $p=0.0075$) and the maximal OD is reduced to 31.2% ($p=0.00001$).

Collectively, these results indicate that while the *vigR* knockdown strain has a reduced growth rate in MH media, it has a specific sensitivity to vancomycin.

Figure 4c: The data presented are not convincing. There is no normalization and quantification on the northern blot. qPCR may be performed to get more precise results. On figure 4c, I cannot find the *vigRdCDS* mutant while it appears in the text.

The ratio of *vigR*:16S rRNA is now quantified and reported and confirms that *vigR* is reduced 86% in the knockdown strain. Quantification on *vigR* levels between JKD6008 and JKD6009 indicate a 21% decrease in JKD6009 and are consistent with additional northern analysis now in Supplementary Figure 3D (see comment immediately below).

Supplementary figure 3c. Quantification is lacking although it is more convincing overnight.

The ratio of *vigR*:16S rRNA is now quantified and reported. *vigR* is increased 20% increase OD 3.0, and 54% increase in overnight cultures in the JKD6008 (VISA) isolate when compared to JKD6009. This quantification is consistent with the *vigR* Northern blot in Figure 4 (lane 1 and 2) and confirms that *vigR* is expressed at a high level in JKD6008 than JKD6009, at mid-log and stationary phase.

Figure 6A: How many replicates were done? As transcript variations are modest, statistical analyses must be performed.

We agree that transcript variations can be modest when comparing between strains and conditions. The Northern blot presented in Figure 6A was used to confirm RNA-seq results which indicated a reduction in *isaA* levels in both the *vigR* deletion condition (FDR=0.00012, log2FC=-1.02) and *vigR* CRISPRi knockdown condition (FDR=0.033, log2FC=-0.69).

Our RNA-seq experiment was performed using three biological replicates in each condition (*vigR* deletion and knockdown condition, total of six biological replicates). The RNA-seq results correlate well with the Northern and GFP fusion quantification results presented in Figure 6A which show an overall reduction in *isaA* transcript levels when the *vigR* 3'UTR is deleted or when *vigR* is knocked down (Table 1 below). We feel that that the multiple orthogonal approaches we have used (RNA-seq, Northern on deletion and knockdown backgrounds, and GFP-fusions) provides convincing evidence that *vigR* increases *isaA* mRNA levels.

Table 1: Relative change in *isaA* expression levels c.f. controls assessed using orthogonal techniques

	RNA-seq	Northern	IsaA-GFP fusion
Δ vigR -3'UTR	-51%	-26%	
vigR CRISPRi	-38%	-20%	
vigR overexpression			+58%

Line 224: 'repression could be partially restored' rather than 'repression could be restored'.

Corrected.

Reviewer #3 (Remarks to the Author):

Review for “RNase III-CLASH of multi-drug resistant *Staphylococcus aureus* reveals a regulatory mRNA 3’UTR required for intermediate vancomycin resistance”, Mediati et al.

Mediati et al. report the identification of a novel regulatory hub in *S. aureus*, VigR, which is involved in the resistance to vancomycin, a last resort antibiotic to treat MRSA. Several interesting new findings are being reported, especially around the unexpected prevalence of mRNA-mRNA interactions detected by RNase III CLASH. Previous research on gram-negative bacteria by the same group used RNase E CLASH to profile sRNA-mRNA interactions associated with endoribonuclease RNase E in pathogenic *E. coli*. Here, the CLASH protocol was adapted (RNase III CLASH) to gram-positive bacteria. Focusing on the clinically important antibiotic resistance of MRSA, the authors identified novel targets of VigR, namely *fold* and *isaA*. The regulatory loop and the functional implications of these RNA interactions have been thoroughly validated using an impressive variety of different techniques and approaches. The finding that an mRNA rather than a protein is crucially involved in regulation of vancomycin resistance is of general interest. The application of dRNA-seq and Term-seq to complement CLASH allows the authors to identify RNA elements within RNA transcript that modulate the expression of CDS as well as novel regulatory sRNAs. Overall, a carefully conducted study with thoroughly validated data revealing interesting new aspects of RNA biology in bacteria.

Points to address:

- The study is a very interesting read but to broaden the readership beyond the bacterial community, a more general introduction into prokaryotic sRNA gene regulation pathways might be useful, also to be able to better understand the role of RNaseIII in the pathway.

Thank you for this suggestion. We have added an introductory sentence on sRNAs with our examples of sRNAs required for antibiotic tolerance (lines 74-76) and have included more information on the role of RNase III in sRNA regulation (lines 85-92).

- CLASH revealed 133 sRNA-mRNA interactions and 543 statistically significant mRNA-mRNA interactions suggesting that mRNAs may exert regulatory functions in trans. Can the authors speculate why in the given experimental setup coding mRNA-mRNA interactions seem to be more prevalent than sRNA-mRNA interactions?

We have been thinking about this during the lockdown and have now published a short opinion piece in mBio (IF=6.78) (Mediati *et al* 2021) discussing why regulatory mRNAs may be more prevalent in bacteria with 5’→3’ exoribonucleases.

We speculate that the abundance of regulatory mRNA interactions in *S. aureus* may be due to the presence of RNase J, an exoribonuclease that degrades RNAs 5’→3’. The model Gram-negatives *E. coli* and *Salmonella* do not have 5’→3’ exoribonuclease activity and regulatory sRNAs processed from 3’UTRs are stable in the absence of inhibitory 5’ structure or 5’PPP.

In *S. aureus*, processed mRNA 3’UTRs can be degraded from the 5’ end by RNase J. We propose that regulatory 3’UTRs are not cleaved in *S. aureus* (forming regulatory mRNA 3’UTRs like *vigR*) to protect the 3’UTR from RNase J attack. We expect that regulatory mRNA interactions may be more prevalent in bacteria that encode 5’→3’ exoribonuclease (RNase J). Additional evidence for this idea is presented in Mediati *et al* mBio 2021 (PMID: 34372700).

The following text has been added into the Discussion (lines 545-553):

“This may be due to the presence of 5’-3’ exoribonuclease activity (RNase J1 and J2) found in Gram-positive *Firmicutes*, which readily degrades free 3’UTR intermediates and may represent an evolutionary barrier to the prevalence of 3’UTR-derived sRNA-mRNA interactions in *S. aureus*.

Regulatory mRNA 3'UTRs like *vigR* that are protected from processing at the 5' end is likely a more widespread regulatory mechanism in *S. aureus* than previously appreciated.”

- It is unclear why replicates 1-2 and 3,4,5,6 underwent different protocols: why 2 samples with one protocol and 4 samples with another and why using different protocols?

We were able to upgrade our UV crosslinking, cell harvesting, and lysis equipment during the course of the study and this meant we needed to update the protocol. The different protocols are described in detail in the Supplementary Methods, and only differ in three steps throughout the RNase III-CLASH protocol.

These differences are:

1. Using the updated Vari-X-linker for crosslinking that allowed faster and lower doses of UV-C (400 mJ) (replicates 1-2) when compared to the W5 small diameter UV-crosslinker (1800 mJ) (replicates 3-6). Both crosslinking units were designed and constructed by UVO3 Ltd, UK.
2. The Vari-X-linker includes cell harvesting by vacuum filtering onto a membrane that allowed faster harvests of large culture volumes (replicates 1-2) when compared to centrifugation (7,000 g, Beckman) (replicates 3-6).
3. We acquired a FastPrep-24 5G instrument (MP Biomedicals) that reduced the time for cell lysis (replicates 1-2) when compared to vortexing (replicates 3-6).

These updates reduced the processing time from sample to cDNA synthesis and gave good correlation of mapped RNA between protocols (Figure 1C).

- In mammalian cells, qPCR is generally performed using 3 reference genes. Here, only 1 was used (*gap*). Is this a particularly stable gene under the conditions applied here or might 3 different reference genes be better?

In our initial qPCR experiments, we tested both the *gap* (SAA6008_RS08745, glyceraldehyde-3-phosphate dehydrogenase) and *pyk* (SAA6008_RS08810, pyruvate kinase) transcripts. The *pyk* transcript was previously used as a qPCR housekeeping gene for *S. aureus* clinical isolates (Theis *et al.* 2007, J. Microbiol. Methods). However, we found that the *gap* transcript was more stably expressed in our clinical MRSA isolate JKD6008 and the CRISPRi pSD1 experiment system, as was shown previously by Sato'o *et al* when using clinical *S. aureus* and the CRISPRi system (Sato'o *et al.* 2018, PLoS One). For this reason, we have used *gap* alone in our experimental setup. Notably, *gap* has been identified as one of the most stably expressed and reference genes across a wide range of conditional bacterial RT-qPCR assays (Cusick *et al.* 2015 PLoS One).

- In the discussion, the authors could elaborate a bit more on the clinical implications their findings might have, i.e. targeting of RNA interactions as a therapeutic approach. Could antisense oligonucleotides be envisaged in the future to target such interactions?

Thank you for the comment and we agree. We have added the following text to the discussion:

“Peptide-conjugated antisense oligonucleotides (ASOs) have gained renewed interest in recent years and allow specific targeting of RNAs for repression. ASOs targeting regulatory RNAs like *vigR* may represent an effective therapeutic approach to re-sensitise VISA isolates to last-line vancomycin treatment.”

- *VigR* 3'UTR is neither independently transcribed nor processed from the *vigR* mRNA. Hence, the authors conclude that *vigR* is a regulatory mRNA and is so far only the 3rd example of bacterial mRNA with trans-regulatory function. Is it possible that *VigR* is not transcribed under the chosen experimental conditions (medium, growth conditions)? It might be interesting to check for *VigR* transcription or processing under different conditions or in other bacterial strains.

We find that the *vigR* 3'UTR is transcribed in our chosen experimental conditions and has regulatory function as an mRNA (ie: without independent transcription or processing). We can't unequivocally rule out that the *vigR* 3'UTR is transcribed or processed under another condition, but we provide evidence that it has regulatory activity as an mRNA UTR.

We have now probed for the *vigR* 3'UTR under the addition growth condition of RPMI-1640 media and find that *vigR* is not independently transcribed or processed under this infection-relevant condition. These results are now presented in Supplementary Figure 3E.

We are currently looking at *vigR* in additional clinical VISA and VSSA isolates and are finding that there is substantial sequence variation in the 3'UTR. These results will be published in a short follow-up publication on the function of *vigR* during infection but falls outside the scope of the current manuscript.

Reviewer comments, second round of review

Reviewer #1 (Remarks to the Author):

The authors have done a great job addressing the reviewers' comments. We recommend the paper be accepted now. For up-to-date literature, there is a new comprehensive review of sRNAs that are processed from bacterial 3'UTRs (An overview of gene regulation in bacteria by small RNAs derived from mRNA 3' ends. FEMS Microbiol Rev. 2022 Apr 7:fuac017. doi: 10.1093/femsre/fuac017, PMID: 35388892) The author may want to include a reference to this paper in lines 451-453.

Reviewer #2 (Remarks to the Author):

In this revised version of the manuscript which is a companion of manuscript submitted by McKellar and collaborators, the authors improved their story and satisfactorily answered to the concerns raised earlier. Using CLASH, they identified numerous interactions. They also used a large variety of techniques. Especially, the authors focused on an unusual 3'UTR. They showed that VigR which is not cleaved (and therefore does not correspond to type I or type II 3'-derived sRNAs) and regulates vancomycin tolerance.

The methods used and the findings obtained are relevant. At this point, I do not have further comments.

Reviewer #3 (Remarks to the Author):

My questions have all been addressed.

Especially, the new paper published by the group in 2021 (Mediati et al., 2021) provides interesting additional information and explanations to their findings presented here. Unless I am mistaken, this paper has, however, not been mentioned in the current manuscript. The authors should add the reference to the relevant section in the discussion.

RESPONSE TO REVIEWERS' COMMENTS

Reviewer #1 (Remarks to the Author):

The authors have done a great job addressing the reviewers' comments. We recommend the paper be accepted now. For up-to-date literature, there is a new comprehensive review of sRNAs that are processed from bacterial 3'UTRs (An overview of gene regulation in bacteria by small RNAs derived from mRNA 3' ends. FEMS Microbiol Rev. 2022 Apr 7:fuac017. doi: 10.1093/femsre/fuac017, PMID: 35388892) The author may want to include a reference to this paper in lines 451-453.

We thank the reviewer for their positive assessment of our revisions and have incorporated the new review article into the manuscript.

Reviewer #2 (Remarks to the Author):

In this revised version of the manuscript which is a companion of manuscript submitted by McKellar and collaborators, the authors improved their story and satisfactorily answered to the concerns raised earlier. Using CLASH, they identified numerous interactions. They also used a large variety of techniques. Especially, they focused on an unusual 3'UTR. They showed that VigR which is not cleaved (and therefore does not correspond to type I or type II 3'-derived sRNAs) and regulates vancomycin tolerance.

The methods used and the findings obtained are relevant. At this point, I do not have further comments.

We thank the reviewer for their positive assessment of our revisions.

Reviewer #3 (Remarks to the Author):

My questions have all been addressed.

Especially, the new paper published by the group in 2021 (Mediati et al., 2021) provides interesting additional information and explanations to their findings presented here. Unless I am mistaken, this paper has, however, not been mentioned in the current manuscript. The authors should add the reference to the relevant section in the discussion.

We thank the reviewer for their positive assessment of our revisions and for highlighting our Opinion piece in mBio. The reviewer is correct that this article contains a more expansive discussion of why regulatory mRNA 3'UTRs may be favoured over regulatory 3'UTR sRNAs in bacteria with 5'→3' exonuclease activity. We have included a reference to our Opinion piece in the discussion (ref 53).